# Tourmaline as a Recorder of Ore-Forming Processes in the Xuebaoding W-Sn-Be Deposit, Sichuan Province, China: Evidence from the Chemical Composition of Tourmaline

**Xinxiang Zhu [1,2], Markus B. Raschke [3] and Yan Liu [2,4,\*]**

1    State Key Laboratory of Geological Processes and Mineral Resources, China University of Geosciences, Beijing 100083, China; 2120180006@cugb.edu.cn
2    Key Laboratory of Deep-Earth Dynamics of Ministry of Natural Resources, Institute of Geology, Chinese Academy of Geological Science, Beijing 100037, China
3    Department of Physics, University of Colorado, Boulder, CO 80303, USA; Markus.raschke@colorado.edu
4    Southern Marine Science and Engineering Guangdong Laboratory (Guangzhou),Ministry of land and resources of China, Geological Survey Bureau, Guangzhou 511458, China
\*    Correspondence: ly@cags.ac.cn

**Abstract:** The Xuebaoding W-Sn-Be deposit located in the Songpan-Ganze Orogenic Belt (Sichuan Province, China) is a hydrothermal deposit with less developed pegmatite stage. The deposit is famous for the coarse-grained crystals of beryl, scheelite, cassiterite, apatite, fluorite, muscovite, and others. The orebody is spatially associated with the Pankou and Pukouling granites hosted in Triassic marbles and schists. The highly fractionated granites are peraluminous, Li-Rb-Cs-rich, and related to W-Sn-Be mineralization. The mineralization can chiefly be classified based on the wallrock and mineral assemblages as muscovite and beryl in granite (Zone I), then beryl, cassiterite and muscovite at the transition from granite to triassic strata (Zone II), and the main mineralized veins composed of an assemblage of beryl, cassiterite, scheelite, fluorite, and apatite hosted in metasedimentary rock units of marble and schist (Zone III). Due to the stability of tourmaline over a wide range of temperature and pressure conditions, its compositional variability can reflect the evolution of the ore-forming fluids. Tourmaline is an important gangue mineral in the Xuebaoding deposit and occurs in the late-magmatic to early-hydrothermal stage, and can thus be used as a proxy for the fluid evolution. Three types of tourmalines can be distinguished: tourmaline disseminations within the granite (type I), tourmaline clusters at the margin of the granite (type II), and tourmalines occurring in the mineralized veins (type III). Based on their chemical composition, both type I and II tourmalines belong to the alkali group and to the dravite-schorl solid solution. Type III tourmaline which is higher in X-site vacancy corresponds to foitite and schorl. It is proposed that the weakly zoned type I tourmalines result from an immiscible boron-rich aqueous fluid in the latest stage of granite crystallization, that the type II tourmalines showing skeletal texture directly formed from the undercooled melts, and that type III tourmalines occurring in the mineralized veins formed directly from the magmatic hydrothermal fluids. Both type I and type II tourmalines show similar compositional variations reflecting the highly fractionated Pankou and Pukouling granites. The higher Ca, Mg, and Fe contents of type III tourmaline are buffered by the composition of the metasedimentary host rocks. The decreasing Na content (<0.8 atoms per formula unit (apfu)) and increasing $Fe^{3+}/Fe^{2+}$ ratios of all tourmaline samples suggest that they precipitated from oxidized, low-salinity fluids. The decreasing trend of Al content from type I (5.60–6.36 apfu) and type II (6.01–6.43 apfu) to type III (5.58–5.87 apfu) tourmalines, and associated decrease in Na, may be caused by the crystallization of albite and muscovite. The combined petrographic, mineralogical, and chemical characteristics of the three types of tourmalines thus reflect the late-magmatic to early-hydrothermal evolution of the ore-forming fluids, and could be used as a geochemical fingerprint for prospecting W-Sn-Be mineralization in the Xuebaoding district.

**Keywords:** tourmaline; geochemistry; W-Sn-Be mineralization; Xuebaoding

## 1. Introduction

Coarse-grained and in part gem-quality beryl, scheelite, cassiterite, apatite, fluorite, tourmaline, muscovite, and other minerals have long been known from an area south of Mount Xuebaoding, 14.5 km northwest of the town of Huya in Pingwu County, Sichuan Province, China. In recent years, the Xuebaoding deposit, primarily associated with two granitic intrusions of Pankou and Pukouling, has gained extensive recognition among mineralogists, economic geologists, and mineral collectors [1–8] for both, producing exceptionally large and aesthetic euhedral crystals, and representing the only-known W, Sn, and Be mineralization in the Songpan-Ganze Orogenic Belt. In previous works, the geological settings, the evolution of ore-forming fluids, and the mineralization with its main minerals such as scheelite and beryl have been studied [2,4–7,9–12]. The Xuebaoding deposit is a rare type of ore deposit, different from those types of ore deposit in that it has W-Sn-Be mineralization together occurring as coarse-grained crystals, and is mainly classified as a hydrothermal deposit rather than pegmatite deposit [4–7].

However, to date no studies have yet examined tourmaline from the Xuebaoding deposit. Tourmaline is a borosilicate mineral with a complex crystal structure and chemical composition, commonly found in igneous, sedimentary, metamorphic and hydrothermal settings [13–28]. Due to its stability over a wide range of temperature and pressure conditions, the composition of tourmaline has been used to address the local environment in which tourmalines have formed, fluid evolution, and ore genesis. In hydrothermal ore deposits, tourmaline composition is controlled by bulk composition of the host rock, pressure, and temperature conditions of the system, and the composition of the hydrothermal fluid [29–35]. Tourmaline is also a common gangue mineral in many ore deposits (e.g., tin, tungsten, beryllium, gold, silver, copper, and uranium) and has been used to characterize the ore-forming processes [9,28,31,32,36–46].

In the Xuebaoding deposit, tourmalines occur in the granites and related W-Sn-Be mineralization where radial tourmaline clusters occur along the granite borders, and are overgrown by muscovite, beryl, quartz, apatite, and other coarse-grained minerals. Locally, needle-like tourmalines of 2–5 cm in length are found as mineral inclusions hosted in quartz, beryl, and albite, within veins. Euhedral tourmaline crystals with diameters of 200–400 μm occur in granitic rocks, in proximity to the ore veins. Crystal chemical analyses of the different types of tourmalines thus allow for the investigation of the relationships between the granites and W-Sn-Be mineralization.

Based on geological field work and sampling carried out in 2005, 2009, 2013, and 2017–2019, thin section petrography, back-scattered electron (BSE) images, and electron microprobe analysis (EMPA), the genesis of tourmalines and the late-magmatic to early-hydrothermal processes are addressed with implications for prospecting W-Sn-Be mineralization.

## 2. Geological Setting

The Xuebaoding deposit is located in the Pankou Dome, north of Longmenshan, and in the western part of the Yangtze Block (Figure 1). The Pukouling and Pankou granites associated with the Xuebaoding deposit intruded into the Triassic metasedimentary rocks along an anticline structure. $^{40}Ar/^{39}Ar$ dating of magmatic muscovite from the Pukouling and Pankou granites gave inverse isochron ages of 200.6 ± 1.2 Ma and 193.7 ± 1.1 Ma, respectively [5]. Recently, five ages for the Xuebaoding deposit have been obtained: (1) $^{40}Ar/^{39}Ar$ dating of quartz-hosted fluid inclusions yielded an age of 191.8 ± 0.7 Ma [2]; (2) $^{40}Ar/^{39}Ar$ plateau age of magmatic muscovite at 189.9 ± 1.8 Ma [3]; (3) Sm/Nd dating of scheelite yielded an isochron age of 182.0 ± 9.2 Ma [4]; (4) U-Pb dating of cassiterite by LA-MC-ICP-MS provided an isochron age of 193.6 ± 6.0 Ma, and the $^{40}Ar/^{39}Ar$ dating of muscovite intergrown with cassiterite yielded an inverse isochron age of 194.53 ± 1.05 Ma [11]. All these ages

indicate that the emplacement of the Pankou and Pukouling granites and the spatially associated mineralization were coeval.

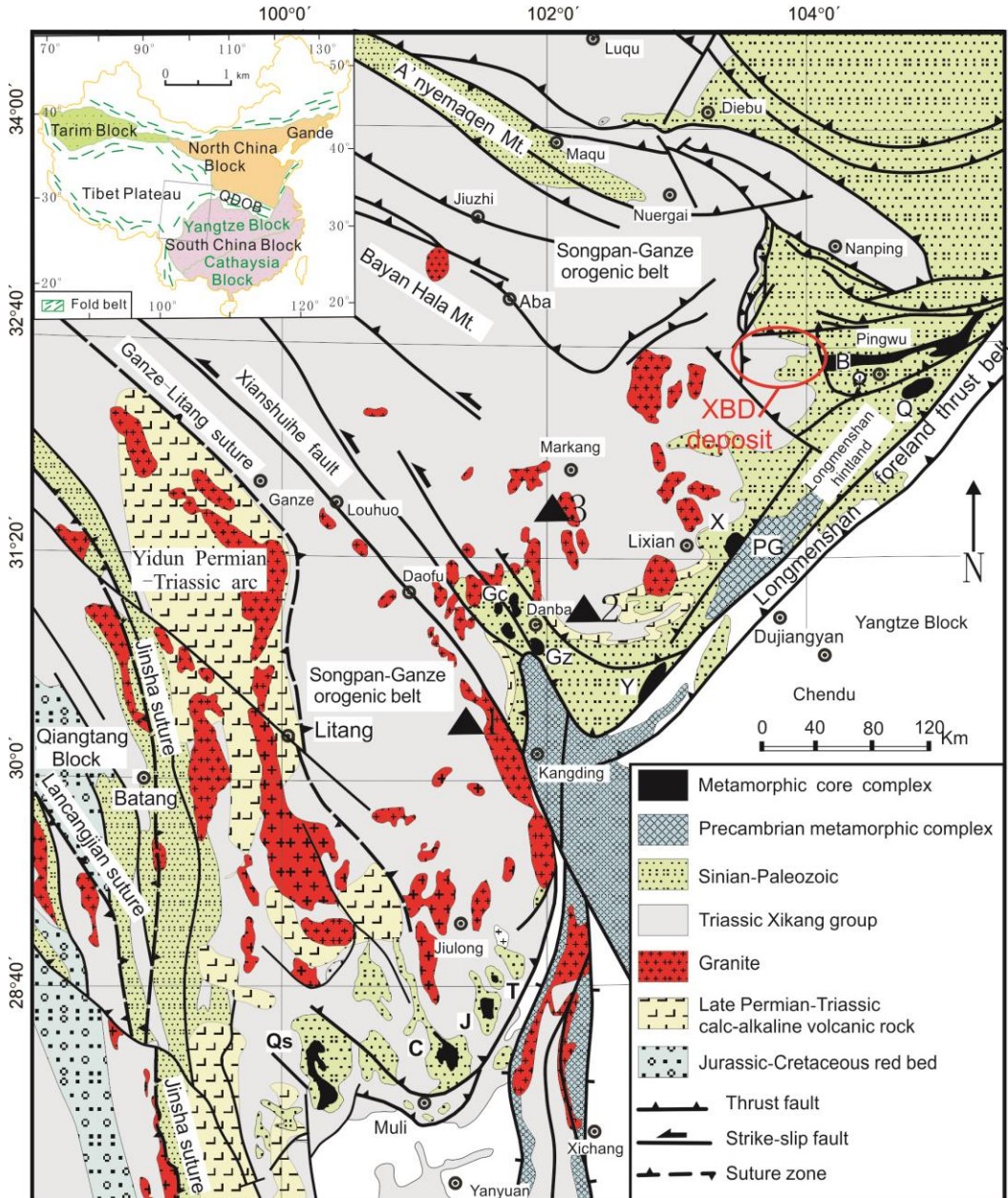

**Figure 1.** Simplified structural geological map of the Songpan-Ganze orogenic belt (map taken from [47]) with Xuebaoding (XBD) deposit location indicated.

The Xuebaoding deposit is hosted in Triassic strata of marble and schist surrounding the two granite intrusions. The mineralization occurs primarily in hydrothermal veins cutting marbles, and breccia bodies and pods in schist. The Xuebaoding deposit is particular for two reasons: (a) in addition to being sparsely disseminated, the crystals of beryl, cassiterite, and scheelite occur in clusters, and in geodes in mineralized veins; (b) the coarse-grained crystals are often euhedral, transparent, large, and of gem-quality.

The Pankou and Pukouling granites are mainly composed of quartz (35%–40%), albite (30%–35%), muscovite (30%–35%), with minor K-feldspar (0%–5%), while mafic minerals are absent. Accessory minerals are zircon, apatite, pyrite, rutile and tourmaline. The Pankou and Pukouling granites are characterized by a central facies and a border facies with a progressively decreasing mineral grain size

towards the border of the granite (Figure 2). The border facies is spatially associated with ore veins, with a high content of ore elements (such as W, Sn, Be) according to whole rock analysis, significantly higher than those of the Clarke [5–7,48–50]. This suggests that the Pankou and Pukouling granites are either the source of the mineralizing fluids or contributed indirectly to the formation of the deposit. In addition, the border facies of the Pankou and Pukouling granites has higher concentrations of $Al_2O_3$ (14.63–21.3 wt.%), $P_2O_5$ (0.07–0.11 wt.%), and B (65–114 ppm) compared to the central facies [6,7]. In term of trace elements, the Pankou and Pukouling granites have low contents of Cr, Ni, Sr, Ba, and Zr, and high contents of Li, Rb, and Cs, typical of highly fractionated granites as described by Wu et al. [51]. Further, with the increasing degree of differentiation, the content of albite in the granites is increasing [5].

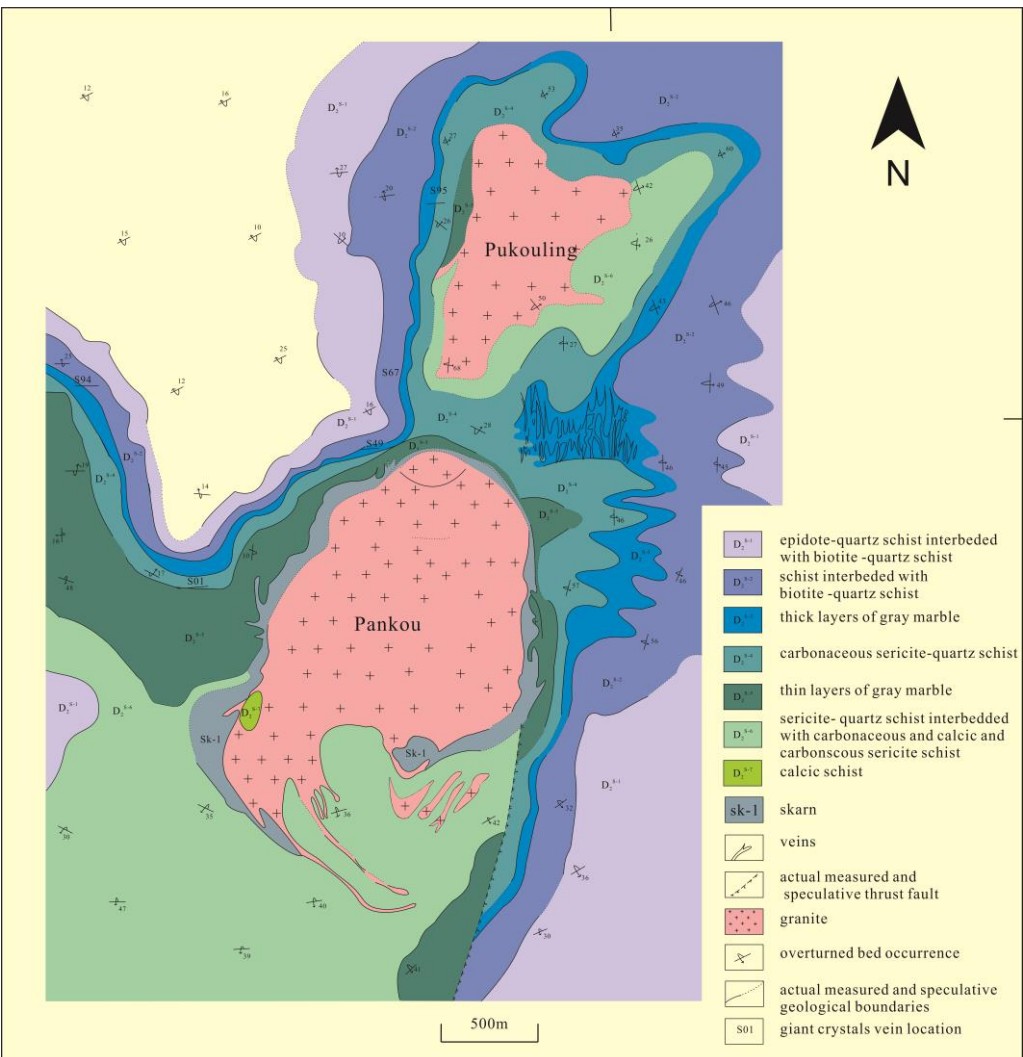

**Figure 2.** Simplified regional geological map of the Pankou and Pukouling granites, with W-Sn-Be mineralization in veins and pods in surrounding upper Triassic marble and schist of the Zhuwo series (modified after reference [52]).

The mineralized veins occur in the extensive joints of the Triassic strata and granites. These consist of a quartz-dominated center, and a margin composed of coarse-grained beryl (0.5–15 cm), cassiterite (1–30 cm), scheelite (1–30 cm), K-feldspar (1–25 cm), albite (1–25 cm), muscovite (1–3 cm), fluorite (1–10 cm), and apatite (0.5–3 cm). This suggests that the monomineralic quartz veins formed after the coarse-grained crystals of the wall. A muscovite fringe separates the mineralized vein from the host rock. Beryl, cassiterite, scheelite, fluorite, apatite, K-feldspar, and albite occur as single crystals or

aggregates overgrown on muscovite. Greisen and other wall rock alterations are not developed in this deposit, and only a very weak muscovite alteration (up to only 0.5 cm thick) and skarn-type alteration were observed in the Xuebaoding deposit.

The crystallization sequence of feldspars in the veins was described by Liu et al. [5]. K-feldspars form at relatively high temperature (500–800 °C), and with decreasing temperature, K-feldspar is replaced by albite on a large scale. Ore minerals such as beryl, scheelite, and cassiterite mostly coexist with albite. In contrast, only beryl, and only in minor amount, is found to coexist with K-feldspar.

Mineralized veins in the Xuebaoding deposit can chiefly be categorized into three zones, based on their mineral assemblages and their host rocks (Figure 3). Zone I is hosted in the granite and is dominated by muscovite, tourmaline, and beryl assemblage. Zone II is mainly composed of beryl, cassiterite, tourmaline, and muscovite and is located at the transition from the granite to metamorphic rock. Zone III represents the main host for the mineralization, with beryl, cassiterite, scheelite, fluorite, calcite, and tourmaline. No crosscutting relationship was observed between the three zones of veins. The calculated H and O isotopic compositions of beryl, scheelite and cassiterite, indicate that the ore-forming fluids are mainly composed of magmatic water with minor meteoric water and $CO_2$ derived from decarbonation of marble [7]. Thus, it is likely that the veins formed during a single hydrothermal stage.

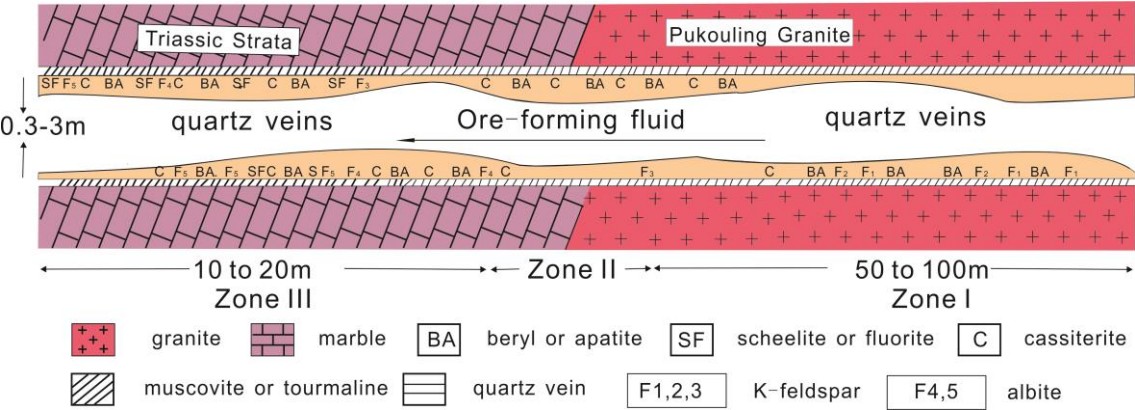

**Figure 3.** Schematic cross-section of a typical mineralized vein showing the distribution of beryl, cassiterite, scheelite, muscovite, tourmaline, albite and K-feldspar and host rock transition from granite to Triassic metamorphic strata and divided into three characteristic zones. After reference [6].

## 3. Sampling and Analytical Methods

### 3.1. Sampling and Petrographic Description

Samples of tourmaline granite, tourmaline clusters and tourmaline inclusions are collected from different locations of the Xuebaoding deposit. Representative samples are shown in Figure 4. Because of the general simple texture of the thin section micrographs, we only provide a qualitative description, followed by quantitative observation of samples.

Based on their occurrences, tourmalines from the Xuebaoding deposit can be divided into three types. Type I tourmaline was collected in the border facies of the granites. It is fine-grained with needles of 200–400 μm in diameter and length of 1–2 mm (Figure 4a). Type I tourmaline crystals are disseminated in the granitic rock, always intergrown with albite and phengite, of euhedral shape, and surrounded by secondary hydrothermal minerals.

Type II tourmalines formed along the margin of the granite as radial clusters. Muscovite layers separate the mineralized vein from the tourmaline clusters (Figure 4b). Locally, tourmaline clusters are overgrown by beryl, albite, scheelite, cassiterite, and quartz (Figure 4b,c). Type II tourmaline crystals are larger than type I with a length up to 0.5–1 cm.

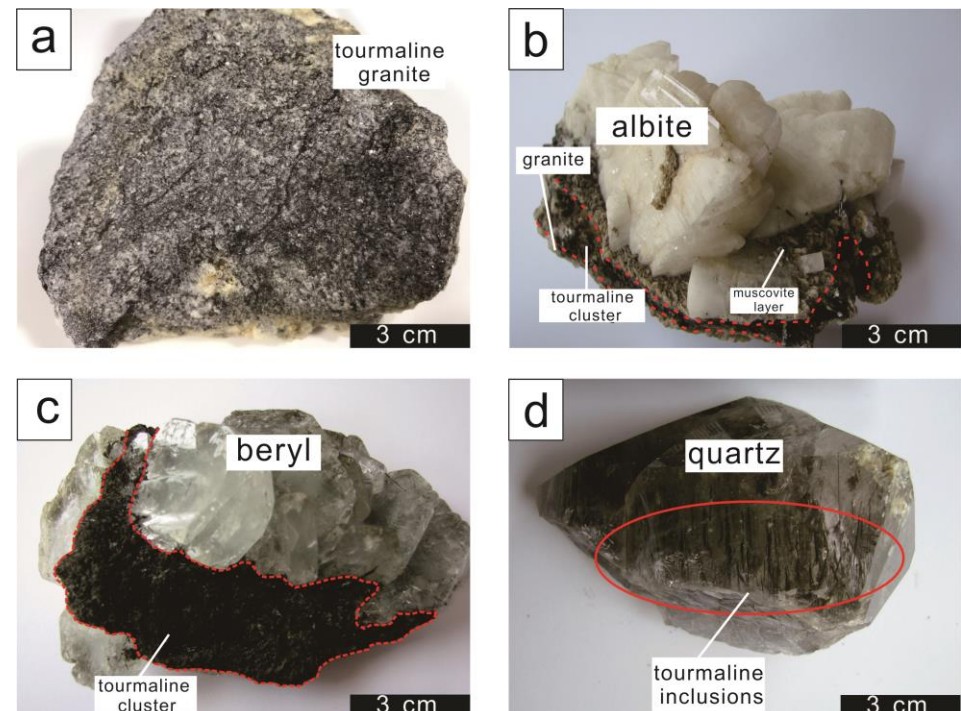

**Figure 4.** (**a**) Tourmaline granite from Xuebaoding, Sichuan Province, China. (**b**) Tourmaline cluster, coarse-graintourmaline and albite from the Xuebaoding, Sichuan Province, China. (**c**) Tourmaline cluster overgrown by tabular beryl in the Xuebaoding deposit, Sichuan Province, China. (**d**) Quartz with tourmaline inclusions from Xuebaoding, Sichuan Province, China.

Type III tourmalines are found as needles in mineralized veins (Figure 4d). The very coarse-grained (2–5 cm) tourmaline crystals always formed as inclusions in other coarse-grained minerals, such as albite, quartz and beryl. Except the size, the difference between type III and type II tourmalines is that type III tourmalines always overgrow on the muscovite layers. Type III tourmaline crystals predominantly show parallel growth (Figure 4d), compared with the radial type II tourmaline.

### 3.2. Analytical Methods

Back scattered electron (BSE) images and chemical compositions of tourmalines were acquired at the Laboratory of Mineralization and Dynamics, Chang'an University (China), using a JXA-8100 electron microprobe analyzer (EMPA), with an acceleration voltage of 15 kV, a beam current of 10 nA, and a spot size less than 10 μm. EMPA standards include the following minerals: andradite for Si and Ca, rutile for Ti, corundum for Al, hematite for Fe, eskolaite for Cr, rhodonite for Mn, bunsenite for Ni, periclase for Mg, albite for Na, and K-feldspar for K.

The general formula of tourmaline is $XY_3Z_6(T_6O_{18})(BO_3)_3VW$ with X = $Na^+$, $K^+$, $Ca^{2+}$, or vacancy; Y = $Fe^{2+}$, $Mg^{2+}$, $Mn^{2+}$, $Al^{3+}$, $Fe^{3+}$, $Cr^{3+}$, $V^{3+}$, $Ti^{4+}$, and $Li^+$; Z = $Al^{3+}$, $Mg^{2+}$, $Fe^{3+}$, $Cr^{3+}$, $V^{3+}$; T = Si, Al; B = B; V = OH, O and W = OH, F, O [18]. Following the normalization procedure of [21], structural formulae of tourmaline were calculated on the basis of a total of 15 cations in the octahedral and tetrahedral (Y + Z + T) sites, all iron was assumed to be $Fe^{2+}$. The proportion of X-site vacancies (❑) was calculated as [1 − (Na + Ca + K)]. F was not analyzed. Results of different crystals are given in Tables 1–3.

**Table 1.** Chemical compositions, determined by EMPA, of type I tourmalines from the Xuebaoding deposit, Sichuan Province, China.

| Sample NO. | XBD 11c | XBD 11d | XBD 11e | XBD 11f | XBD 11g | XBD 21a | XBD 21b | XBD 21c | XBD 21d | XBD 21f | XBD 21h | XBD 21i | XBD 21j | XBD 21k | XBD 21l | XBD 21m | XBD 21o | XBD 21p | XBD 21q | XBD 21r | XBD 21s | XBD 21t | XBD 22a |
|---|---|---|---|---|---|---|---|---|---|---|---|---|---|---|---|---|---|---|---|---|---|---|---|
| $SiO_2$ | 35.64 | 35.9 | 35.24 | 36.27 | 35.48 | 36.92 | 36.98 | 36.63 | 37.1 | 37 | 37.09 | 36.14 | 36.67 | 35.53 | 35.83 | 37.24 | 36.92 | 36.53 | 36.78 | 36.58 | 36.53 | 35.99 | 36.19 |
| $TiO_2$ | 0.63 | 1.12 | 0.72 | 0.97 | 1.27 | 0.35 | 0.43 | 0.29 | 0.42 | 0.41 | 0.78 | 0.72 | 0.4 | 0.82 | 0.76 | 0.2 | 0.69 | 0.5 | 0.42 | 0.33 | 0.4 | 0.87 | 0.75 |
| $Al_2O_3$ | 32.36 | 32.52 | 32.27 | 32.34 | 32.13 | 34.07 | 33.15 | 33.88 | 31.52 | 31.19 | 32.16 | 33.08 | 33.42 | 32.37 | 32.32 | 33.88 | 32.62 | 33.35 | 33.5 | 31.59 | 32.43 | 33.39 | 32.29 |
| $MgO$ | 4.6 | 3.78 | 4.64 | 5.68 | 4.07 | 3.07 | 3.14 | 2.76 | 5.03 | 4.93 | 4.19 | 3.67 | 3.42 | 3.9 | 3.7 | 4.07 | 4.13 | 3.83 | 3.79 | 4.89 | 4.38 | 2.84 | 3.04 |
| $MnO$ | 0.4 | 0.38 | 0.39 | 0.3 | 0.4 | 0.4 | 0.45 | 0.46 | 0.2 | 0.17 | 0.37 | 0.33 | 0.31 | 0.34 | 0.41 | 0.09 | 0.42 | 0.35 | 0.41 | 0.15 | 0.19 | 0.44 | 0.43 |
| $FeO$ | 9.19 | 9 | 8.91 | 7.4 | 9.62 | 10.35 | 10.3 | 10.72 | 9.95 | 10.03 | 9.4 | 10 | 9.05 | 9.47 | 9.6 | 8.16 | 8.93 | 9.07 | 9.47 | 9.57 | 10.3 | 10.42 | 10.07 |
| $Cr_2O_3$ | 0 | 0 | 0 | 0.02 | 0 | 0.01 | 0.01 | 0.03 | 0 | 0.02 | 0 | 0.02 | 0.05 | 0 | 0 | 0 | 0 | 0 | 0.01 | 0 | 0 | 0.02 | 0.03 |
| $CaO$ | 1.08 | 1.15 | 1.16 | 1.25 | 1.15 | 0.37 | 0.65 | 0.37 | 0.76 | 0.72 | 1.07 | 0.92 | 0.44 | 1.18 | 0.98 | 0.06 | 1.15 | 0.79 | 0.55 | 0.7 | 1.01 | 0.92 | 0.82 |
| $Na_2O$ | 2.15 | 2.14 | 2.14 | 2.07 | 2.07 | 1.84 | 2.34 | 1.73 | 2.4 | 2.4 | 2.04 | 2.28 | 1.77 | 2.01 | 2.01 | 2.08 | 2.39 | 1.98 | 2.04 | 2.5 | 2.18 | 2.18 | 2.18 |
| $K_2O$ | 0.04 | 0.03 | 0.02 | 0.03 | 0.04 | 0.02 | 0.07 | 0.02 | 0.05 | 0.05 | 0.04 | 0.05 | 0.02 | 0.06 | 0.03 | 0.03 | 0.06 | 0.04 | 0.05 | 0.05 | 0.04 | 0.06 | 0.04 |
| Total | 86.1 | 86.01 | 85.49 | 86.33 | 86.23 | 87.39 | 87.51 | 86.9 | 87.43 | 86.92 | 87.13 | 87.21 | 85.55 | 85.68 | 85.62 | 85.8 | 87.31 | 86.42 | 87.02 | 86.36 | 87.47 | 87.12 | 85.84 |
| Si | 5.66 | 5.74 | 5.63 | 5.7 | 5.66 | 5.84 | 5.82 | 5.85 | 5.8 | 5.82 | 5.85 | 5.69 | 5.9 | 5.7 | 5.76 | 5.92 | 5.78 | 5.8 | 5.81 | 5.77 | 5.73 | 5.71 | 5.82 |
| Al(total) | 6.05 | 6.11 | 6.07 | 5.98 | 6.03 | 6.34 | 6.14 | 6.36 | 5.79 | 5.77 | 5.97 | 6.13 | 6.33 | 6.11 | 6.11 | 6.34 | 6.01 | 6.23 | 6.22 | 5.86 | 5.98 | 6.23 | 6.11 |
| Al(T) | 0.05 | 0.11 | 0.07 | 0 | 0.03 | 0.16 | 0.14 | 0.15 | 0 | 0 | 0 | 0.13 | 0.1 | 0.11 | 0.11 | 0.08 | 0.01 | 0.2 | 0.19 | 0 | 0 | 0.23 | 0.11 |
| Al(Z) | 6 | 6 | 6 | 5.98 | 6 | 6 | 6 | 6 | 5.79 | 5.77 | 5.97 | 6 | 6 | 6 | 6 | 6 | 6 | 6 | 6 | 5.86 | 5.98 | 6 | 6 |
| Al(Y) | 0 | 0 | 0 | 0 | 0 | 0.18 | 0 | 0.21 | 0 | 0 | 0 | 0 | 0.23 | 0 | 0 | 0.26 | 0 | 0.03 | 0.03 | 0 | 0 | 0 | 0 |
| Ti | 0.08 | 0.13 | 0.09 | 0.11 | 0.15 | 0.04 | 0.05 | 0.03 | 0.05 | 0.05 | 0.09 | 0.09 | 0.05 | 0.1 | 0.09 | 0.02 | 0.08 | 0.06 | 0.05 | 0.04 | 0.05 | 0.1 | 0.09 |
| Mg | 1.1 | 0.91 | 1.11 | 1.34 | 0.97 | 0.73 | 0.74 | 0.66 | 1.18 | 1.16 | 0.99 | 0.87 | 0.83 | 0.94 | 0.89 | 0.97 | 0.97 | 0.91 | 0.9 | 1.16 | 1.03 | 0.68 | 0.73 |
| Mn | 0.05 | 0.05 | 0.05 | 0.04 | 0.05 | 0.05 | 0.06 | 0.06 | 0.03 | 0.02 | 0.05 | 0.04 | 0.04 | 0.05 | 0.06 | 0.01 | 0.06 | 0.05 | 0.06 | 0.02 | 0.03 | 0.06 | 0.06 |
| Fe(total) | 1.22 | 1.2 | 1.19 | 0.97 | 1.28 | 1.36 | 1.35 | 1.43 | 1.3 | 1.31 | 1.24 | 1.31 | 1.21 | 1.27 | 1.29 | 1.08 | 1.17 | 1.2 | 1.25 | 1.26 | 1.35 | 1.38 | 1.35 |
| Cr | 0 | 0 | 0 | 0 | 0 | 0 | 0 | 0 | 0 | 0 | 0 | 0 | 0.01 | 0 | 0 | 0 | 0 | 0 | 0 | 0 | 0 | 0 | 0 |
| Ca | 0.18 | 0.2 | 0.2 | 0.21 | 0.2 | 0.06 | 0.11 | 0.06 | 0.13 | 0.12 | 0.18 | 0.15 | 0.08 | 0.2 | 0.17 | 0.01 | 0.19 | 0.13 | 0.09 | 0.12 | 0.17 | 0.16 | 0.14 |
| Na | 0.66 | 0.66 | 0.66 | 0.63 | 0.64 | 0.56 | 0.71 | 0.53 | 0.72 | 0.73 | 0.62 | 0.69 | 0.55 | 0.62 | 0.62 | 0.64 | 0.72 | 0.61 | 0.62 | 0.76 | 0.66 | 0.67 | 0.68 |
| K | 0.01 | 0.01 | 0 | 0.01 | 0.01 | 0 | 0.01 | 0 | 0.01 | 0.01 | 0.01 | 0.01 | 0 | 0.01 | 0.01 | 0.01 | 0.01 | 0.01 | 0.01 | 0.01 | 0.01 | 0.01 | 0.01 |
| X-site vacancy | 0.15 | 0.14 | 0.13 | 0.15 | 0.15 | 0.37 | 0.16 | 0.4 | 0.14 | 0.14 | 0.19 | 0.14 | 0.37 | 0.16 | 0.2 | 0.35 | 0.07 | 0.25 | 0.27 | 0.11 | 0.16 | 0.16 | 0.17 |
| Species name | Dravite | Dravite | Dravite | Schorl | Dravite | Dravite | Dravite | Dravite | Dravite | Dravite | Dravite | Dravite | Dravite | Dravite | Dravite | Dravite | Dravite | Dravite | Dravite | Dravite | Dravite | Dravite | Dravite |

| Sample NO. | XBD 22b | XBD 22c | XBD 22e | XBD 22f | XBD 22g | XBD 22h | XBD 22e | XBD 22f | XBD 22g | XBD 22h | XBD 22i | XBD 22j | XBD 22k | XBD 34a | XBD 34b | XBD 34c | XBD 34d | XBD 34e | XBD 34f | XBD 34g | XBD 34h | XBD 34i | XBD 34 |
|---|---|---|---|---|---|---|---|---|---|---|---|---|---|---|---|---|---|---|---|---|---|---|---|
| $SiO_2$ | 37.15 | 36.85 | 35.4 | 37.31 | 36.87 | 37.21 | 35.4 | 37.31 | 36.87 | 37.21 | 36.02 | 37.24 | 37.31 | 36.88 | 36.09 | 36.24 | 36.14 | 36.53 | 36.73 | 36.77 | 36.67 | 36.49 | 36.79 |
| $TiO_2$ | 0.23 | 0.28 | 0.73 | 0.3 | 0.37 | 0.3 | 0.73 | 0.3 | 0.37 | 0.3 | 0.29 | 0.27 | 0.24 | 0.78 | 0.72 | 0.41 | 0.78 | 0.75 | 0.91 | 0.73 | 0.54 | 0.5 | 0.84 |
| $Al_2O_3$ | 33.27 | 33.46 | 31.93 | 33.82 | 33.35 | 33.16 | 31.93 | 33.82 | 33.35 | 33.16 | 32.64 | 33.94 | 34.25 | 32.85 | 32.51 | 32.43 | 32.35 | 32.43 | 32.21 | 30.15 | 30.5 | 30.67 | 30.31 |
| $MgO$ | 4.51 | 3.43 | 2.98 | 3.88 | 3.8 | 3.84 | 2.98 | 3.88 | 3.8 | 3.84 | 3.71 | 3.94 | 3.7 | 2.71 | 2.81 | 2.71 | 3.59 | 3.67 | 4.66 | 5.32 | 5.33 | 5.43 | 5.52 |
| $MnO$ | 0.31 | 0.35 | 0.5 | 0.38 | 0.3 | 0.44 | 0.5 | 0.38 | 0.3 | 0.44 | 0.69 | 0.36 | 0.32 | 0.46 | 0.42 | 0.43 | 0.41 | 0.4 | 0.32 | 0.12 | 0.15 | 0.11 | 0.11 |
| $FeO$ | 8.01 | 9.34 | 10.08 | 8.14 | 8.85 | 8.8 | 10.08 | 8.14 | 8.85 | 8.8 | 9.17 | 8.94 | 9.3 | 10.72 | 10 | 10.74 | 10.15 | 10.02 | 8.3 | 9.84 | 9.47 | 9.84 | 9.77 |
| $Cr_2O_3$ | 0.05 | 0.02 | 0 | 0.01 | 0.05 | 0.1 | 0 | 0.01 | 0.05 | 0.1 | 0.07 | 0.03 | 0.02 | 0 | 0 | 0.01 | 0.05 | 0.05 | 0.01 | 0 | 0.02 | 0.01 | 0 |
| $CaO$ | 0.34 | 0.55 | 0.87 | 0.37 | 0.39 | 0.54 | 0.87 | 0.37 | 0.39 | 0.54 | 0.5 | 0.36 | 0.39 | 0.78 | 0.92 | 0.8 | 1.01 | 1.06 | 1.2 | 1.53 | 1.34 | 1.48 | 1.63 |
| $Na_2O$ | 1.8 | 1.95 | 2.04 | 1.85 | 1.99 | 1.93 | 2.04 | 1.85 | 1.99 | 1.93 | 1.77 | 1.76 | 1.84 | 2.09 | 2.19 | 2.13 | 2.06 | 2.03 | 2.12 | 2.08 | 2.23 | 2.14 | 2.15 |
| $K_2O$ | 0.03 | 0.01 | 0.03 | 0.03 | 0.04 | 0.05 | 0.03 | 0.03 | 0.04 | 0.05 | 0.05 | 0.04 | 0.03 | 0.03 | 0.05 | 0.06 | 0.04 | 0.05 | 0.03 | 0.04 | 0.05 | 0.04 | 0.03 |
| Total | 85.71 | 86.23 | 84.55 | 86.09 | 86.01 | 86.37 | 84.55 | 86.09 | 86.01 | 86.37 | 84.9 | 86.86 | 87.39 | 87.29 | 85.69 | 85.92 | 86.55 | 86.99 | 86.51 | 86.58 | 86.28 | 86.72 | 87.15 |
| Si | 5.92 | 5.88 | 5.79 | 5.94 | 5.88 | 5.91 | 5.79 | 5.94 | 5.88 | 5.91 | 5.84 | 5.89 | 5.87 | 5.86 | 5.82 | 5.84 | 5.76 | 5.79 | 5.8 | 5.82 | 5.8 | 5.75 | 5.78 |
| Al(total) | 6.24 | 6.28 | 6.14 | 6.33 | 6.26 | 6.2 | 6.14 | 6.33 | 6.26 | 6.2 | 6.23 | 6.31 | 6.34 | 6.14 | 6.17 | 6.15 | 6.07 | 6.05 | 5.99 | 5.61 | 5.68 | 5.69 | 5.6 |
| Al(T) | 0.08 | 0.12 | 0.14 | 0.06 | 0.12 | 0.09 | 0.14 | 0.06 | 0.12 | 0.09 | 0.16 | 0.11 | 0.13 | 0.14 | 0.17 | 0.15 | 0.07 | 0.05 | 0 | 0 | 0 | 0 | 0 |
| Al(Z) | 6 | 6 | 6 | 6 | 6 | 6 | 6 | 6 | 6 | 6 | 6 | 6 | 6 | 6 | 6 | 6 | 6 | 6 | 5.99 | 5.61 | 5.68 | 5.69 | 5.6 |
| Al(Y) | 0.16 | 0.16 | 0 | 0.27 | 0.14 | 0.12 | 0 | 0.27 | 0.14 | 0.12 | 0.07 | 0.2 | 0.21 | 0 | 0 | 0 | 0 | 0 | 0 | 0 | 0 | 0 | 0 |
| Ti | 0.03 | 0.03 | 0.09 | 0.04 | 0.04 | 0.04 | 0.09 | 0.04 | 0.04 | 0.04 | 0.04 | 0.03 | 0.03 | 0.09 | 0.09 | 0.05 | 0.09 | 0.09 | 0.11 | 0.09 | 0.06 | 0.06 | 0.1 |
| Mg | 1.08 | 0.82 | 0.73 | 0.93 | 0.91 | 0.92 | 0.73 | 0.93 | 0.91 | 0.92 | 0.9 | 0.94 | 0.87 | 0.65 | 0.68 | 0.66 | 0.86 | 0.87 | 1.1 | 1.26 | 1.27 | 1.28 | 1.3 |
| Mn | 0.04 | 0.05 | 0.07 | 0.05 | 0.04 | 0.06 | 0.07 | 0.05 | 0.04 | 0.06 | 0.1 | 0.05 | 0.04 | 0.06 | 0.06 | 0.06 | 0.06 | 0.05 | 0.04 | 0.02 | 0.02 | 0.02 | 0.01 |
| Fe(total) | 1.06 | 1.24 | 1.37 | 1.08 | 1.18 | 1.17 | 1.37 | 1.08 | 1.18 | 1.17 | 1.24 | 1.18 | 1.22 | 1.42 | 1.34 | 1.44 | 1.35 | 1.32 | 1.09 | 1.3 | 1.25 | 1.29 | 1.28 |
| Cr | 0.01 | 0 | 0 | 0 | 0.01 | 0.01 | 0 | 0 | 0.01 | 0.01 | 0.01 | 0 | 0 | 0 | 0 | 0 | 0 | 0.01 | 0 | 0 | 0 | 0 | 0 |
| Ca | 0.06 | 0.09 | 0.15 | 0.06 | 0.07 | 0.09 | 0.15 | 0.06 | 0.07 | 0.09 | 0.09 | 0.06 | 0.07 | 0.13 | 0.16 | 0.14 | 0.17 | 0.18 | 0.2 | 0.26 | 0.23 | 0.25 | 0.27 |
| Na | 0.56 | 0.6 | 0.65 | 0.57 | 0.61 | 0.59 | 0.65 | 0.57 | 0.61 | 0.59 | 0.56 | 0.54 | 0.56 | 0.64 | 0.68 | 0.66 | 0.64 | 0.62 | 0.65 | 0.64 | 0.68 | 0.65 | 0.65 |
| K | 0.01 | 0 | 0.01 | 0.01 | 0.01 | 0.01 | 0.01 | 0.01 | 0.01 | 0.01 | 0.01 | 0.01 | 0 | 0.01 | 0.01 | 0.01 | 0.01 | 0.01 | 0.01 | 0.01 | 0.01 | 0.01 | 0.01 |
| X-site vacancy | 0.38 | 0.3 | 0.2 | 0.36 | 0.31 | 0.3 | 0.2 | 0.36 | 0.31 | 0.3 | 0.35 | 0.4 | 0.37 | 0.22 | 0.15 | 0.19 | 0.18 | 0.19 | 0.14 | 0.09 | 0.08 | 0.09 | 0.07 |
| Species name | Schorl | Dravite | Dravite | Dravite | Dravite | Dravite | Dravite | Dravite | Dravite | Dravite | Dravite | Dravite | Dravite | Dravite | Dravite | Dravite | Dravite | Dravite | Schorl | Dravite | Schorl | Dravite | Schorl |

**Table 2.** Chemical compositions, determined by EMPA, of type II tourmalines from the Xuebaoding deposit, Sichuan Province, China.

| Sample NO. | TUR 72a | TUR 72b | TUR 72c | TUR 72d | TUR 72e | TUR 72f | TUR 74a | TUR 74b | TUR 74c | TUR 74d | TUR 74e | TUR 74f | TUR 74g | TUR 74h | TUR 74i | TUR 74j | TUR 74k |
|---|---|---|---|---|---|---|---|---|---|---|---|---|---|---|---|---|---|
| $SiO_2$ | 36.39 | 36.55 | 36.83 | 36.22 | 36.46 | 36.53 | 36.47 | 36.47 | 36.38 | 36.52 | 35.69 | 36.70 | 35.80 | 36.92 | 36.72 | 36.58 | 36.25 |
| $TiO_2$ | 0.36 | 0.70 | 0.53 | 0.64 | 0.80 | 0.47 | 0.33 | 0.22 | 0.16 | 0.21 | 0.22 | 0.33 | 0.41 | 0.27 | 0.18 | 0.31 | 0.48 |
| $Al_2O_3$ | 33.67 | 32.73 | 33.32 | 32.18 | 32.16 | 32.68 | 33.67 | 33.99 | 34.20 | 33.96 | 33.49 | 33.49 | 32.74 | 33.82 | 33.99 | 33.62 | 32.65 |
| MgO | 3.47 | 4.56 | 4.23 | 4.61 | 4.23 | 4.60 | 3.86 | 4.02 | 3.69 | 3.50 | 3.69 | 4.40 | 4.85 | 4.54 | 3.74 | 3.74 | 4.72 |
| MnO | 0.16 | 0.15 | 0.12 | 0.13 | 0.15 | 0.15 | 0.14 | 0.14 | 0.13 | 0.14 | 0.09 | 0.11 | 0.15 | 0.14 | 0.11 | 0.10 | 0.13 |
| FeO | 10.03 | 8.41 | 8.93 | 9.09 | 8.83 | 8.22 | 9.22 | 9.20 | 9.48 | 9.65 | 8.85 | 8.89 | 8.14 | 8.59 | 9.74 | 9.32 | 8.10 |
| $Cr_2O_3$ | 1.10 | 0.03 | 0.04 | 0.01 | 0.00 | 0.05 | 0.02 | 0.06 | 0.00 | 0.00 | 0.01 | 0.00 | 0.07 | 0.02 | 0.03 | 0.00 | 0.01 |
| CaO | 0.33 | 1.01 | 0.81 | 0.98 | 1.01 | 0.99 | 0.47 | 0.27 | 0.29 | 0.37 | 0.38 | 0.42 | 0.71 | 0.40 | 0.35 | 0.38 | 0.96 |
| $Na_2O$ | 1.48 | 2.05 | 2.12 | 2.17 | 2.03 | 2.06 | 1.87 | 1.91 | 1.64 | 1.78 | 1.88 | 2.10 | 1.84 | 2.06 | 1.97 | 2.00 | 2.18 |
| $K_2O$ | 0.02 | 0.01 | 0.04 | 0.04 | 0.03 | 0.02 | 0.03 | 0.00 | 0.03 | 0.01 | 0.02 | 0.03 | 0.03 | 0.01 | 0.01 | 0.03 | 0.02 |
| Total | 87.00 | 86.19 | 86.96 | 86.05 | 85.69 | 85.75 | 86.07 | 86.27 | 85.98 | 86.14 | 84.31 | 86.46 | 84.74 | 86.77 | 86.84 | 86.07 | 85.48 |
| Si | 5.80 | 5.79 | 5.79 | 5.75 | 5.83 | 5.81 | 5.81 | 5.79 | 5.82 | 5.83 | 5.80 | 5.80 | 5.76 | 5.80 | 5.80 | 5.83 | 5.77 |
| Al(total) | 6.32 | 6.10 | 6.17 | 6.01 | 6.05 | 6.11 | 6.32 | 6.35 | 6.43 | 6.38 | 6.40 | 6.22 | 6.20 | 6.25 | 6.32 | 6.30 | 6.11 |
| Al(T) | 0.20 | 0.10 | 0.17 | 0.01 | 0.05 | 0.11 | 0.19 | 0.21 | 0.18 | 0.17 | 0.20 | 0.20 | 0.20 | 0.20 | 0.20 | 0.17 | 0.11 |
| Al(Z) | 6.00 | 6.00 | 6.00 | 6.00 | 6.00 | 6.00 | 6.00 | 6.00 | 6.00 | 6.00 | 6.00 | 6.00 | 6.00 | 6.00 | 6.00 | 6.00 | 6.00 |
| Al(Y) | 0.12 | 0.00 | 0.00 | 0.00 | 0.00 | 0.00 | 0.13 | 0.14 | 0.25 | 0.22 | 0.20 | 0.02 | 0.00 | 0.06 | 0.13 | 0.13 | 0.00 |
| Ti | 0.04 | 0.08 | 0.06 | 0.08 | 0.10 | 0.06 | 0.04 | 0.03 | 0.02 | 0.03 | 0.03 | 0.04 | 0.05 | 0.03 | 0.02 | 0.04 | 0.06 |
| Mg | 0.83 | 1.08 | 1.00 | 1.10 | 1.01 | 1.10 | 0.92 | 0.96 | 0.89 | 0.84 | 0.90 | 1.04 | 1.17 | 1.07 | 0.89 | 0.89 | 1.13 |
| Mn | 0.02 | 0.02 | 0.02 | 0.02 | 0.02 | 0.02 | 0.02 | 0.02 | 0.02 | 0.02 | 0.01 | 0.01 | 0.02 | 0.02 | 0.02 | 0.01 | 0.02 |
| Fe(total) | 1.33 | 1.11 | 1.17 | 1.20 | 1.18 | 1.09 | 1.22 | 1.22 | 1.26 | 1.28 | 1.20 | 1.17 | 1.09 | 1.12 | 1.28 | 1.24 | 1.07 |
| Cr | 0.14 | 0.00 | 0.00 | 0.00 | 0.00 | 0.01 | 0.00 | 0.01 | 0.00 | 0.00 | 0.00 | 0.00 | 0.01 | 0.00 | 0.00 | 0.00 | 0.00 |
| Ca | 0.06 | 0.17 | 0.14 | 0.17 | 0.17 | 0.17 | 0.08 | 0.05 | 0.05 | 0.06 | 0.07 | 0.07 | 0.12 | 0.07 | 0.06 | 0.06 | 0.16 |
| Na | 0.46 | 0.63 | 0.64 | 0.67 | 0.63 | 0.63 | 0.58 | 0.59 | 0.51 | 0.55 | 0.59 | 0.64 | 0.57 | 0.63 | 0.60 | 0.62 | 0.67 |
| K | 0.00 | 0.00 | 0.01 | 0.01 | 0.01 | 0.00 | 0.01 | 0.00 | 0.01 | 0.00 | 0.00 | 0.01 | 0.01 | 0.00 | 0.00 | 0.01 | 0.00 |
| X-site vacancy | 0.48 | 0.20 | 0.21 | 0.16 | 0.19 | 0.19 | 0.34 | 0.37 | 0.44 | 0.38 | 0.34 | 0.28 | 0.30 | 0.30 | 0.34 | 0.31 | 0.16 |
| Species name | Dravite | Dravite | Dravite | Dravite | Dravite | Schorl | Dravite | Dravite | Dravite | Dravite | Dravite | Dravite | Schorl | Dravite | Dravite | Dravite | Schorl |

**Table 3.** Chemical compositions, determined by EMPA, of type III tourmalines from the Xuebaoding deposit, Sichuan Province, China.

| Sample NO. | XBDTur2 | XBDTur3 | XBDTur4 | XBDTur5 | XBDTur6 | XBDTur7 | XBDTur8 | XBDTur11 |
|---|---|---|---|---|---|---|---|---|
| $SiO_2$ | 37.90 | 38.03 | 37.93 | 37.84 | 38.20 | 37.28 | 37.23 | 37.91 |
| $TiO_2$ | 0.33 | 0.28 | 0.04 | 0.93 | 0.16 | 0.45 | 0.96 | 0.97 |
| $Al_2O_3$ | 32.99 | 33.05 | 32.77 | 32.38 | 32.91 | 30.18 | 30.92 | 31.49 |
| MgO | 3.67 | 3.72 | 3.67 | 4.45 | 3.94 | 4.73 | 4.38 | 4.46 |
| MnO | 0.51 | 0.31 | 0.45 | 0.16 | 0.37 | 0.26 | 0.35 | 0.36 |
| FeO | 8.83 | 8.93 | 8.60 | 8.52 | 9.32 | 7.94 | 8.25 | 8.41 |
| $Cr_2O_3$ | 0.00 | 0.15 | 0.06 | 0.00 | 0.01 | 0.08 | 0.03 | 0.03 |
| CaO | 0.24 | 0.26 | 0.31 | 1.18 | 0.17 | 1.09 | 1.31 | 1.34 |
| $Na_2O$ | 1.30 | 1.66 | 1.48 | 1.66 | 1.70 | 1.94 | 1.80 | 1.83 |
| $K_2O$ | 0.05 | 0.02 | 0.03 | 0.06 | 0.04 | 0.12 | 0.04 | 0.04 |
| Total | 85.84 | 86.40 | 85.33 | 87.17 | 86.82 | 84.06 | 85.28 | 86.85 |
| Si | 5.92 | 5.93 | 5.99 | 5.92 | 5.91 | 6.08 | 5.97 | 5.97 |
| Al(total) | 5.85 | 5.84 | 5.87 | 5.74 | 5.78 | 5.58 | 5.62 | 5.62 |
| Al(T) | 0.00 | 0.00 | 0.00 | 0.00 | 0.00 | 0.00 | 0.00 | 0.00 |
| Al(Z) | 5.85 | 5.84 | 5.87 | 5.74 | 5.78 | 5.58 | 5.62 | 5.62 |
| Al(Y) | 0.00 | 0.00 | 0.00 | 0.00 | 0.00 | 0.00 | 0.00 | 0.00 |
| Ti | 0.07 | 0.06 | 0.01 | 0.19 | 0.03 | 0.09 | 0.20 | 0.20 |
| Mg | 0.74 | 0.75 | 0.74 | 0.89 | 0.78 | 0.99 | 0.90 | 0.90 |
| Mn | 0.13 | 0.08 | 0.12 | 0.04 | 0.10 | 0.07 | 0.09 | 0.09 |
| Fe(total) | 2.30 | 2.32 | 2.26 | 2.22 | 2.40 | 2.16 | 2.21 | 2.21 |
| Cr | 0.00 | 0.03 | 0.01 | 0.00 | 0.00 | 0.02 | 0.01 | 0.01 |
| Ca | 0.06 | 0.06 | 0.07 | 0.28 | 0.04 | 0.27 | 0.32 | 0.32 |
| Na | 0.32 | 0.41 | 0.37 | 0.41 | 0.42 | 0.50 | 0.46 | 0.46 |
| K | 0.01 | 0.00 | 0.01 | 0.02 | 0.01 | 0.03 | 0.01 | 0.01 |
| X-site vacancy | 0.60 | 0.52 | 0.55 | 0.29 | 0.53 | 0.19 | 0.21 | 0.21 |
| Species name | Foitite | Foitite | Foitite | Schorl | Foitite | Schorl | Schorl | Schorl |

## 4. Results

### 4.1. Textural Features Of Tourmaline

Type I tourmalines occurs as euhedral grains varying from 200 to 300 μm in diameter on BSE images (Figure 5a,b). They are poorly zoned with dark cores surrounded by bright rims. The abundances of feldspar in the interstitial areas between type I tourmaline grains in the tourmaline-bearing granite show some differences between the Pankou and Pukouling granites. The K-feldspar content is lower, and albite is more abundant (Figure 5a–c), compared to the border facies of the granite (Figure 2). Muscovite is transformed into phengite on a large scale. Biotite does not occur in tourmaline-bearing granite. Further, scheelite forms grains of about 10 μm in size and generally occurring in fractures of phengite (Figure 5a).

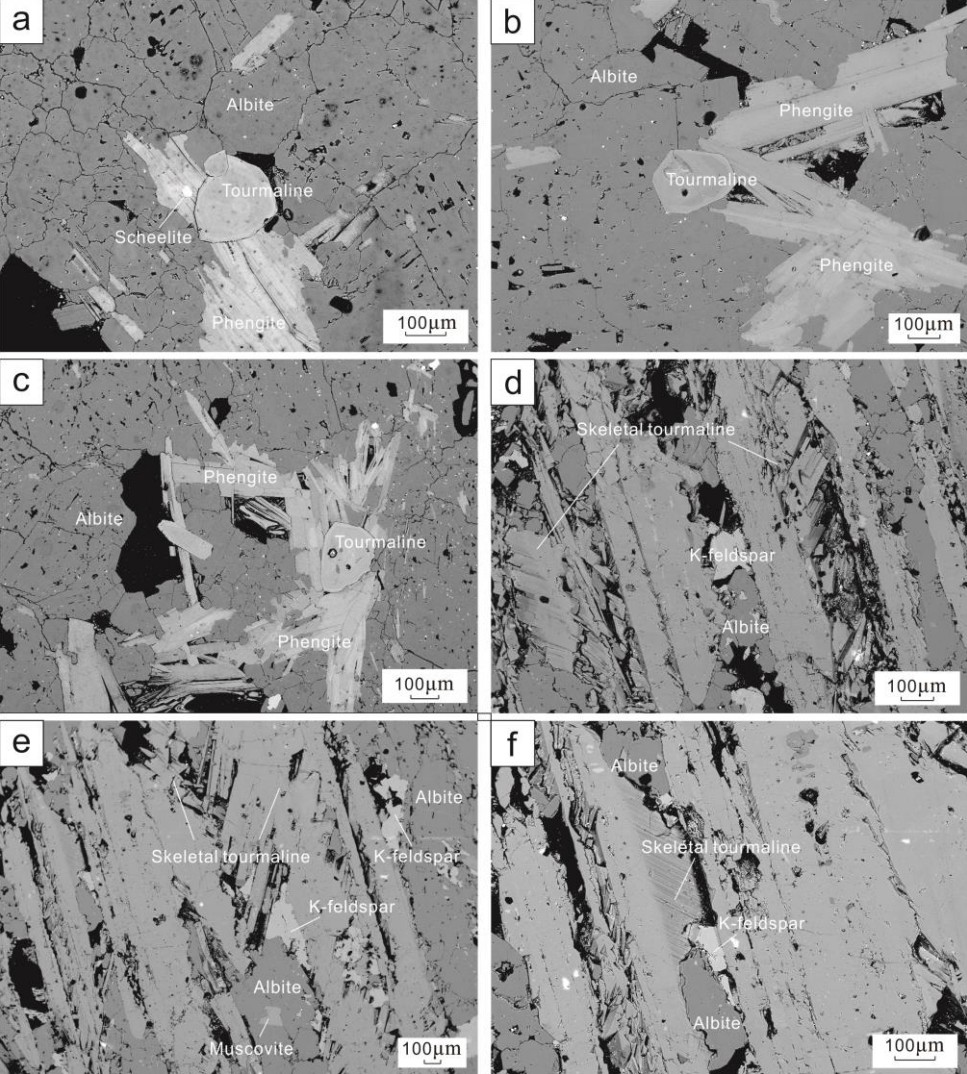

**Figure 5.** Backscattered electron images of tourmaline from the Xuebaoding deposit. (**a**) Type I: weakly zoned tourmaline intergrown with phengite and albite with fine-grained scheelite nearby, in the granite. (**b**,**c**) Type I: weakly zoned tourmaline intergrown with phengite and albite in the granite. (**d**–**f**) Type II: radial clusters of skeletal tourmaline hosted in ore vein.

Type II tourmalines occur as radial clusters along the margin of the granite. No distinct compositional domains or zoning are observed within single tourmaline crystals on the BSE images (Figure 5d). In particular, the larger tourmalines show a skeletal texture which are composed of

numerous small (50–150 μm) disconnected fine-grained tourmaline crystals (Figure 5e,f). Type II tourmaline crystals show characteristics of hydrothermal alteration on the BSE images (Figure 5). Tourmaline crystals are 1–2 cm in size and always coexist with albite, K-feldspar, and muscovite (Figure 5e). K-feldspar is more abundant at the margin of the granite, with large scale albitization. (Figure 5d,e). Muscovite here has a relatively smaller size compared with the phengite coexisting with type I tourmaline (Figure 5e).

The very coarse-grain type III tourmalines (2–5 cm) occur in the mineralized veins. They are always found as inclusions within albite, quartz, and beryl in the mineralized veins. K-feldspar is not found to coexist with this type of tourmaline (Figure 4d). Sometimes, type III tourmaline crystals are found to grow above the muscovite layer, which can distinguish the two types (Figure 4b).

## 4.2. Chemical Composition of Tourmaline

Type I: The chemical compositions of tourmaline show variations of Si (5.63–5.94 apfu), $Al_{total}$ (5.60–6.36 apfu), Ti (0.02–0.15 apfu), Mg (0.65–1.34 apfu), $Fe_{total}$ (0.97–1.44 apfu), Ca (0.01–0.27 apfu), Na (0.53–0.76 apfu) and X-site vacancy (0.07–0.40 apfu) (Table 1). The content of Mg, Fe, Na, Al and X-site vacancy seem to vary greatly. Color variations respond to increasing Ca, Ti, Fe, Mg, and X-site vacancies and decreasing Al towards brighter rims. In the Al-Fe-Mg ternary diagram after reference [20], tourmaline compositions plot in the field of Li-poor granitoid and their associated pegmatites, aplites, metapelites, and psammites (Figure 6a). In the Fe-Mg-Ca ternary diagram, type I tourmaline compositions plot in the field of Ca-poor metapelites, psammites, and calc-silicate rocks, and the near Li-poor granitoid and their associated pegmatites and aplites (Figure 6b). All tourmalines belong to the alkali group (Figure 6c) and are mostly plot dominantly in the dravite/Oxy-dravite fields, while a few tourmaline plot in the schorl/Oxy-schorl field (Figure 6d). Together with the average $Al_{total}$ value (5.98 apfu) nearly to 6 apfu (Figure 7a) and the (Fe + Mg) values (2.01–2.58 apfu) higher than 2 apfu (Figure 7b), type I tourmaline belongs to the dravite-schorl series ($NaMg_3Al_6Si_6O_{18}(BO_3)_3(OH)_3OH$ and $NaFe^{2+}_3Al_6Si_6O_{18}(BO_3)_3(OH)_3OH$) which have a higher content of X-site vacancy [22].

Type II: Tourmalines show compositional variations of Si (5.75–5.83 apfu), $Al_{total}$ (6.01–6.43 apfu), Ti (0.02–0.10 apfu), Mg (0.83–1.17 apfu), $Fe_{total}$ (1.07–1.33 apfu), Ca (0.05–0.17 apfu), Na (0.46–0.67 apfu) and X-site vacancy (0.16–0.48 apfu) (Table 2). The Al values of type II tourmalines are relatively higher than that of type I, and the Na contents are relatively lower. In the Al-Fe-Mg and the Fe-Mg-Ca ternary diagrams (Figure 6a,b), type II and type I tourmaline compositions closely overlap with each other. Type II tourmaline belongs to the alkali group (Figure 6c), and plot at the limit of the schorl/Oxy-schorl and dravite/Oxy-dravite fields (Figure 6d). With the content of $Al_{total}$ (6.01–6.43 apfu) nearly to 6 apfu (Figure 7a) and the (Fe + Mg) values (2.10–2.32 apfu) nearly to 3 apfu, type II could also be classified as dravite-schorl which have a higher content of X-site vacancy and Al than common dravite and schorl. The low (Fe + Mg) values of type II tourmalines are similar to type I tourmalines implying a low Mg and Fe fluids.

Type III: The chemical compositions of tourmaline show variations of Si (5.91–6.08 apfu), $Al_{total}$ (5.58–5.87 apfu), Ti (0.01–0.20 apfu), Mg (0.74–0.90 apfu), $Fe_{total}$ (2.16–2.40 apfu), Ca (0.06–0.32 apfu), Na (0.32–0.50 apfu), and X-site vacancy (0.19–0.60 apfu) (Table 3). The contents of Na and Al are obviously lower than type I and II tourmalines. The Fe, Mg, Ca and X-site vacancy values are higher than that in type I and II tourmalines. In the Fe-Mg-Ca ternary diagram (Figure 6a), type III tourmaline compositions plot in the field of $Fe^{3+}$-rich quartz-tourmaline rocks. Tourmaline compositions are plotted in the Li-rich granitoid, pegmatites, aplites and Li-poor granitoids, pegmatites, aplites in the Fe-Mg-Ca ternary diagrams (Figure 6b). Due to the variation of X-site vacancy, Type III tourmaline belongs to the alkali group and vacancy group (Figure 6c) and plot at the limit of the schorl/Oxy-schorl and foitite/❏-Fe-O root name fields (Figure 6d). With the high content of (Fe + Mg) values more than 3 apfu and the $Al_{total}$ (5.58–5.87 apfu) nearly to 6 apfu, type III could be classified as schorl ($NaFe^{2+}_3Al_6Si_6O_{18}(BO_3)_3(OH)_3OH$) which have a higher content of X-site vacancy and foitite (❏

$(Fe^{2+}{}_2Al)Al_6Si_6O_{18}(BO_3)_3(OH)_3OH$ ) whose Al content is relatively lower. The high (Fe + Mg) values of type III tourmalines imply a relatively higher Mg and Fe content in fluids.

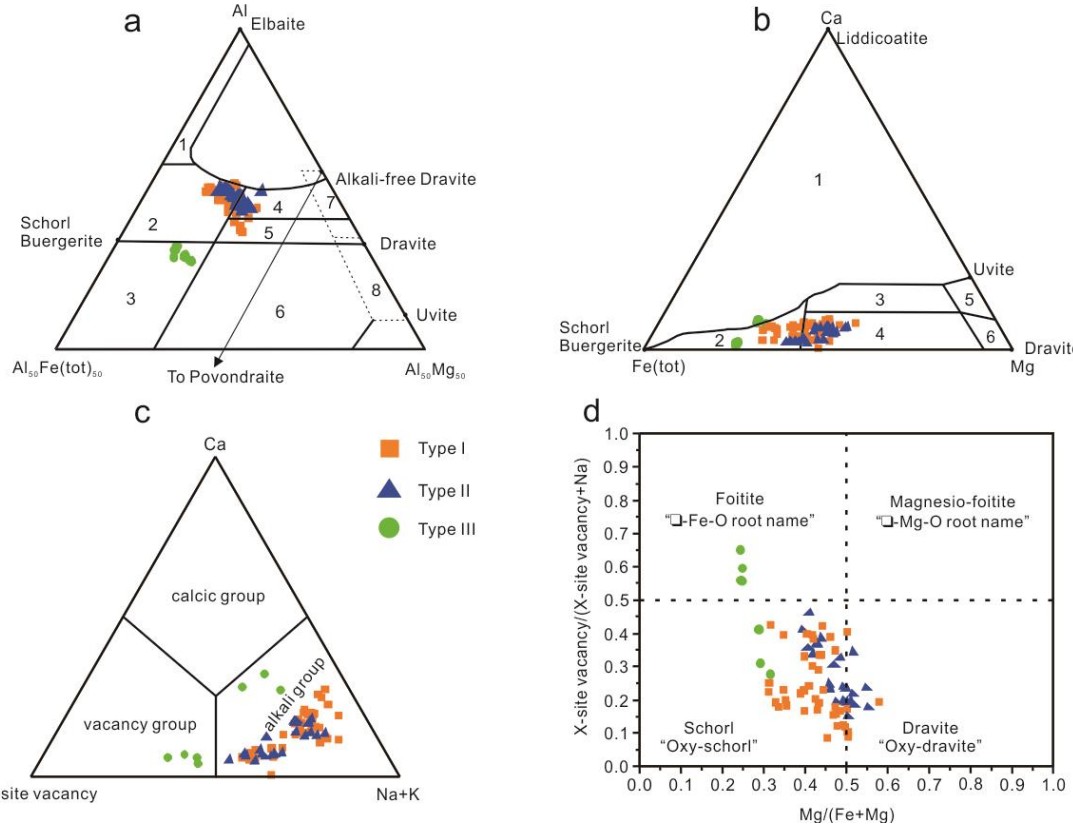

**Figure 6.** Ternary and binary diagrams showing the chemical composition of tourmaline from the Xuebaoding deposit. (**a**) Al-Fe-Mg ternary diagram after reference [20]. The fields represent typical tourmaline compositions from: (1) Li-rich granitoid, pegmatites, aplites; (2) Li-poor granitoids, pegmatites, aplites; (3) $Fe^{3+}$-rich quartz-tourmaline rocks; (4) metapelites and metapsammites with an Al-saturating phase; (5) metapelites and metapsammites without an Alsaturating phase; (6) $Fe^{3+}$-rich quartz-tourmaline rocks, calc-silicate rocks, and metapelites; (7) low-Ca mmeta-ultramafics and Cr, V-rich metasediments; and (8) metacarbonates and meta-pyroxenites. (**b**) Ca-Fe-Mg ternary diagram after reference [20]. These fields are: (l) Li-rich granitoid pegmatites and aplites; (2) Li-poor granitoids and associated pegmatites and aplites; (O), (3) Ca-rich metapelites, metapsammites, and calc-silicate rocks; (4) Ca-poor metapelites, metapsammites, and quartz-tourmaline rocks; (5) Metacarbonates, and (6) Metaultramafics. (**c**) Ternary classification diagram of tourmaline subgroup based on the occupancy of the X-site (modified after reference [23]). (**d**) Plot of Mg/(Fe + Mg) vs. X-site vacancy/(Na + X-site vacancy) of tourmaline grains (modified after reference [24]).

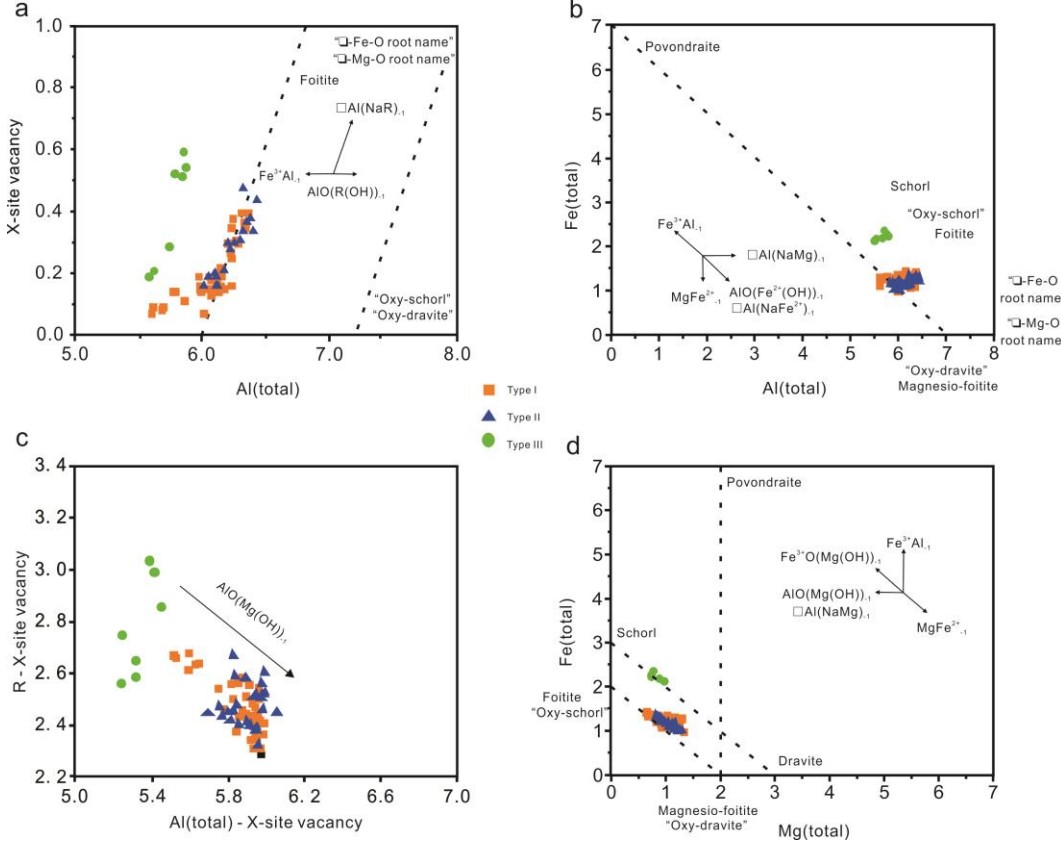

**Figure 7.** Binary diagrams showing the main substitutions in tourmaline. (**a**) Al(total) vs. X-site vacancy plot. (**b**) Al(total) vs. Fe(total) plot. (**c**) Al(total) – X-site vacancy vs. R – X-site vacancy plot. (**d**) Mg(total) vs. Fe(total) plot. Diagrams after reference [24]. R = Mg + Fe + Mn. The directions of selected exchange vectors are shown for reference.

## 5. Discussion

### 5.1. Origin of Tourmaline

In previous studies, three hypotheses have been proposed for the origin of tourmaline in granitic rocks: (1) Crystallization can occur from a boron-rich granitic melt [26,33]; (2) Crystallization proceeds from an immiscible boron-rich hydrous melt or fluid that segregated in the late-magmatic stage [13,29,35,44]; lastly, (3) post-magmatic hydrothermal alteration of the granite by an externally supplied boron-rich fluid [29,34]. In Pankou and Pukouling granites, the initially formed K-feldspar is largely replaced by albite during the evolution of the highly fractionated magma, hence, only few K-feldspars are preserved [5]. Type I tourmalines occur as disseminated euhedral crystals in the granites and coexist with albite and phengite (Figure 5a–c), and no K-feldspar is found. In the highly fractionated granite, the abundance of K-feldspar decrease with the increasing degree of differentiation. Phengite is generally regarded as the alteration product of muscovite. In addition, fine-grained scheelite is found coexisting with type I tourmaline (Figure 5a).

Tourmaline formed from hydrothermal fluids shows fine-scale, oscillatory-type zoning [33,35,42]. Euhedral type I tourmaline crystals are poorly zoned with dark cores surrounded by bright rims. Evidence above implies that these types may not crystallize directly from a B-rich melt even in granites.

Chemical compositions of the Pankou and Pukouling granites as well as the intruded marble were investigated by previous studies [5–7]. The Pankou and Pukouling granites are characterized by high contents of $Al_2O_3$ (14.63–21.30 wt.%) and B (65–114 ppm), but have low contents of CaO (0.57–0.74 wt.%), MgO (0.26–0.4 wt.%), and $Fe_2O_3$ (0.33–1.27 wt.%). Chemical compositions of type I tourmalines are similar to the Pankou and Pukouling granites with a relatively high content of

$Al_{total}$ (5.60–6.36 apfu) and B, and low contents of Ca (0.01–0.27 apfu), Mg (0.65–1.34 apfu), and $Fe_{total}$ (0.97–1.44 apfu). Previous studies [7] show that the ore-forming fluids are mainly composed of magmatic water with minor meteoric water and $CO_2$ derived from decarbonation of marble. Thus, these tourmalines seem not to have crystallized from post-magmatic hydrothermal metasomatism by infiltrating boron-rich fluid.

Type I tourmalines are generally intergrowth with albite and phengite. In highly fractionated magma, albite alway replace K-feldspar with the increasing degree of differentiation. And phengite is generally regarded as the product of hydrothermal alteration. While the tourmaline crystals are texturally isolated within the granite, there are also no joints or fissures around these crystals. In addition, according to previous fluid inclusion studies [4], liquid immiscibility has been proposed in the Pankou and Pukouling melts during its evolution. Hence, because of the mixture signature of magmatic and hydrothermal, it is likely that type I tourmalines are products of an immiscible boron-rich aqueous fluid released during the late-magmatic to hydrothermal transition, as observed in many magmatic-hydrothermal deposits elsewhere [13,29,33,35,37,42,44].

Like most of the magmatic tourmalines, type II tourmalines exhibit no zoning [26,33,37]. Notably type II tourmalines show a skeletal texture (Figure 5d–f) similar to the Stone Mountain tourmalines described by Longfellow and Swanson [33], and the occurrence of skeletal tourmaline crystals is cited as evidence of undercooled crystallization of magma. These authors proposed a model for skeletal tourmaline formation: the crystallization of the melt starts along the margins, tourmalines nucleate and grow from a highly fractionated and undercooled melt, which is B-rich, resulting in skeletal crystals; crystallization of skeletal tourmaline along the margins deplete B and raises the solidus temperature, resulting in crystallization at that margin and the remaining fluids then crystallize at lower temperatures, and produce euhedral tourmaline and coarse-grained minerals.

Type II tourmaline, which is skeletal, always grows along the margin of the granite. Border facies of Pankou and Pukouling granites show compositional features of highly fractionated granites, and aplite is generally thought as the product of such a granitic magma [51]. Thus, the border facies of the Pankou and Pukouling granites may closely connect with aplite. Radial tourmaline clusters are often overgrown by coarse-grained muscovite, beryl, and albite (Figure 4b,c). The skeletal type II tourmalines coexisting with muscovite, albite and K-feldspar (Figure 5d–f). The K-feldspar are obviously a replacement remnant. Mica here is mainly muscovite and is relatively less than that coexisting with type I tourmalines. It is likely that the B-rich melt intrude into triassic strata and crystallized from the margin; B and Al of the melt firstly supply the growth of tourmaline which caused the relatively small size of muscovite. Due to the low degree of crystal differentiation in the margin, K-feldspar is not completely replaced by albite. In addition, type II tourmalines generally reflect the compositions of the host rocks with the high $Al_{total}$ (6.01–6.43 apfu) and low Mg (0.83–1.17 apfu), $Fe_{total}$ (1.07–1.33 apfu) and Ca (0.05–0.17 apfu) values. As proposed above, it is most likely that type II tourmaline clusters form directly from the melts in the late-magmatic stage. Furthermore, the remaining fluids crystallized at lower undercooling, which may produce euhedral type I tourmalines found in granites and tourmaline inclusions of other coarse-grained minerals found in the veins.

Type III tourmaline occurring in the veins is always separated by muscovite layers from the granite margin and type II tourmaline clusters. These tourmalines are always found as inclusions hosted in beryl, albite, K-feldspar and quartz (Figure 4d). Type III tourmaline crystals are larger than types I and II. Abundance of K-feldspar decrease with decreasing temperature, in the mineralized veins. Albite could be found intergrowing with all the ore minerals (beryl, scheelite and cassirtite), while K-feldspar only coexisting with minor beryl. Thus, type III tourmalines seem to have crystallized in the early hydrothermal process (crystallizing later than muscovite, and earlier than K-feldspar and other coarse-grain minerals). Chemical compositions further support this model with the content of Mg (0.74–0.90 apfu), $Fe_{total}$ (2.16–2.40 apfu), and Ca (0.06–0.32 apfu) higher and the $Al_{total}$ (5.58–5.87 apfu) values relatively lower than that of type I and II tourmalines. These high Mg and Fe values are inconsistent with the Pankou and Pukouling leucogranite, thus it may be caused by the interaction

between hydrothermal fluid and biotite schist (Figure 2). With the growth sequence, mineralogical characteristics, and chemical composition variations of type I, II, and III tourmalines, this can effetiely target the evolution during late-magmatic to early-hydrothermal stage of fluids

Based on the evidences above, we can deduce the genesis of two types of tourmalines. At first, highly fractionated Pankou and Pukouling magma rise and intrude into triassic strata causing some joints and fissures. During this process, skeletal tourmalines (type II) form from the undercooled and B-rich melt and gradually grow into radial clusters along the margin of highly fractionated granites. Then, type I tourmalines, as suggested products of an immiscible boron-rich aqueous fluid, form in the remaining fluids. Ore-forming fluids flow into fissures and joints of the metamorphic strata, lastly forming the type III tourmalines and other coarse-grained minerals.

*5.2. Chemical Evolution of Tourmaline*

Tourmaline is the most important borosilicate mineral because of its ubiquity and the diversity of petrologic information that it can yield. Once formed, it does not readily readjust its composition by volume diffusion, even at relatively high temperatures [18–24]. This chemically complex borosilicate is mechanically and chemically refractory, found in many rock types, and stable over a wide range of geological conditions [13,14,16,17,26,27]. During its formation, tourmaline is sensitive to its chemical environment and responds to chemical changes in coexisting minerals and fluids, activities of $H_2O$ and dissolved species, and pressure and temperature conditions [29–35]. As demonstrated in Section 5.1 above, variations of tourmaline compositions may record the late-magmatic to early-hydrothermal transition.

The chemical compositions of types I and II are similar and overlap in the variation diagrams (Figures 6 and 7). In the Al-Fe-Mg diagram (Figure 6a), chemical compositions of type I and II tourmalines plot in the field of Li-poor granitoid and their associated pegmatites and aplites and metapelites and psammites. In the Fe-Mg-Ca ternary diagram (Figure 6b), the compositions of type I and II tourmalines fall in the field of Ca-poor metapelites, psammites, and calc-silicate rocks, and the near Li-poor granitoid and their associated pegmatites and aplites. Compositions of type II tourmalines are similar to type I, implying a relatively similar crystallization environment with type II to type I tourmalines. The low Ca content of tourmaline is somewhat unexpected for the Xuebaoding deposit, because the ore veins are mainly hosted in the Ca-rich metamorphic strata. It is likely that the tourmaline composition was buffered by the composition of the Pankou and Pukouling granites which are low in Ca. For these leucogranites, tourmaline is the major mafic mineral, which leads to the schorl-dravite species. In addition, as pointed out by Wu at al. [51], tourmalines usually show compositional variations from the early Mg- and Fe-bearing to later Al-bearing elbaite, in highly fractionated magmas. Thus, the relatively high $Al_{total}$ (nearly 6 apfu) and low Mg + Fe (<3 apfu) values of type I and II tourmalines could further provide their late-magmatic origin.

Type III tourmaline compositions plot in the field of $Fe^{3+}$-rich quartz-tourmaline rocks, in the Fe-Mg-Ca ternary diagram (Figure 6a), and the Li-rich granitoid, pegmatites, aplites, and Li-poor granitoids, pegmatites, aplites in the Fe-Mg-Ca ternary diagrams (Figure 6b). The concentrations of Al, Mg, Fe, Ca, and Na in type III tourmaline vary significantly, compared to type I and II tourmalines. It is likely that the increasing Mg + Fe (>3 apfu) and Ca (0.06–0.32 apfu) values are caused by the reaction between fluid and triassic strata (biotite schist and marble), and the decreasing of Al and Na values are caused by the crystallization of albite and muscovite.

Type I and II tourmalines from the Xuebaoding deposit share similar characteristics. In the $Al_{total}$ vs. X-site vacancy diagram (Figure 7a), compositions of type I and II tourmalines show positive correlation between $Al_{total}$ and X-site vacancies. Linear regression of the data gives two positive slopes, implying that the variations of Al, Mg, and Na are mainly due to the ☐ Al(NaR)$_{-1}$ substitution vector with influence by others. In the $Al_{total}$ vs. $Fe_{total}$ (Figure 7b) diagram, tourmaline compositions plot almost along the ☐Al(NaMg)$_{-1}$ vector. In the Al(total) − X-site vacancy vs. R − X-site vacancy diagram (Figure 7c), compositional data of tourmaline is discrete. In the Mg(total) vs. Fe(total) diagram

(Figure 7d), compositional data of tourmaline show a relatively steep negative slope close to $-1.0$. With the (Fe + Mg) values less than 3 apfu and Al values nearly to 6 apfu, the contribution of the $Fe^{3+}Al_{-1}$ vector is ruled out. Hence, it is likely that the Al, Mg, and Na variations of type I and I tourmalines are explained mostly by the ❑ $Al(NaMg)_{-1}$ and $MgFe^{2+}_{-1}$ vectors. Al values of type III show highly positive correlation with X-site vacancy (Figure 7a), and hence, the dominant substitution is probably ❑ $Al(NaMg)_{-1}$ with some influence of $Fe^{3+}Al_{-1}$ vector. In the $Al_{total}$ vs. $Fe_{total}$ (Figure 7b) diagram, the relatively steeper positive slopes also indicates the influence of $Fe^{3+}Al_{-1}$ vector. Similar to type I and II tourmalines, compositional data of type III tourmaline is discrete in the Al(total) − X-site vacancy vs. R − X-site vacancy diagram (Figure 7c), and plot close to the ❑ $Al(NaMg)_{-1}$ vector in Figure 7d. Differently, (Fe + Mg) values of type III tourmalines are higher than 3 apfu (Figure 7d) and the content of Al is lower (average in 5.73 < 6 apfu), which may indicate the contribution of the $Fe^{3+}Al_{-1}$ vector in type III tourmalines. Hence, it is likely that the dominant substitution of type III tourmalines are ❑ $Al(NaMg)_{-1}$ and $MgFe^{2+}_{-1}$ vectors with some contributions of $Fe^{3+}Al_{-1}$ vector.

Usually, chemical compositions of $Fe^{3+}$-rich tourmalines generally follows a trend between "oxy-dravite" $[Na(Mg_2Al)(Al6)(Si_6O_{18})(BO_3)_3(OH)_3(O)]$ and povondraite $[Na(Fe_3^{3+})(Fe_4^{3+}Mg_2)(Si_6O_{18})(BO_3)_3(OH)_3(O)]$ (e.g., [40,53]). In our study, type I and II tourmalines do not follow this trend (Figure 7c). Furthermore, Type I and type II tourmalines show similar substitution mechanisms, mainly caused by ❑$Al(NaMg)_{-1}$ and $MgFe^{2+}_{-1}$ vectors in the substitution diagrams (Figure 7). While influence of $Fe^{3+}Al_{-1}$ vector is found with type III tourmalines (Figure 7), implying a increasing trend of $Fe^{3+}/Fe^{2+}$ ratios. This suggests that the tourmalines at Xuebaoding precipitated from a gradually oxidized fluids. The relatively low Na values (<0.76 apfu) of all tourmalines suggest that they precipitated from fluids of low- to moderate-salinity.

### 5.3. Implication for W-Sn-Be Mineralization

The Xuebaoding deposit is a hydrothermal deposit with less developed pegmatite stage, shown by the H and O isotopic analysis, and the well-preserved crystal shape of coarse-grained minerals [4–7]. This is consistent with the occurrence of the three type tourmalines, W-Sn-Be mineralization is mainly related with the hydrothermal type III tourmaline, and only minor scheelite could be found coexisting with type I tourmaline (Figure 5a). The chemical compositions of type I, type II, and type III tourmaline could respond to the Na and Al decreasing or Ca, Mg, and Fe increasing characteristics of the ore-forming fluid during late-magmatic to early-hydrothermal stage. The low Na and increasing $Fe^{3+}/Fe^{2+}$ ratios of three types of tourmalines imply a fluid of low- to moderate-salinity and gradually oxidized condition. In addition, skarn is found around the Xuebaoding deposit, with the widely distributed calcite, fluorite, apatite and tourmaline, implying the enrichment of volatile compounds such as B, F, $CO_2$, P, Cl, and Ca from marble. Such characteristics of ore-forming fluids can be a reason, explaining why W-Sn-Be mineralization mainly occurs mainly in hydrothermal stage

The Be content of undifferentiated granite is approximately 4–6 ppm, which could not result in the crystallization of beryl, according to Wu et al. [51]. Be commonly forms chloride complexes and fluoride (mainly fluoride ) complexes in hydrothermal fluids [54,55]. In addition, the existence of Al stabilizes complexes in fluid solution [54]. As discussed above, the Pankou and Pukouling granites are highly peraluminous, while Al content of fluid decrease during late-magmatic to early-hydrothermal evolution process. Under this condition, beryllium complexes decompose and Be-bearing minerals (mainly beryl in Xuebaoding) start to crystallize during early-hydrothermal evolution. Furthermore, a large amount of fluorite and calcite found in the Zone III of ore veins, suggests a decreasing $F^-$ and $Cl^-$ content in the fluid, which could favor the crystallization of beryl. Generally, in alkali chloride-bearing, acid solutions, Sn transportation is affected by a complex series of stannous chloride-bearing species, including: simple chloride, mixed ligand (hydroxy chloride), and alkali-bearing chloride and hydroxy chloride stannous complexes ($SnOHCl$, $SnCl_2$, $KSnOHCl_2$, $KSnCl_3$, $K_2SnCl_4$, $K_3SnCl_5$, $K_4SnCl_6$ and $NaSnOHCl_2$, $NaSnCl_3$, etc) [56–58]. Taylor et al. [56] suggested that the solubility of $SnO_2$ is limited by decreasing fluid acidity (mainly $HCl_{(aq)}$) and oxidized conditions, with little influence by temperature.

In the Xuebaoding deposit, the high $Fe^{3+}/Fe^{2+}$ ratios of type III tourmaline imply a oxidized condition during hydrothermal stage of fluids, which is beneficial to precipitation of $SnO_2$. The Sn-rich fluid flow into Triassic strata through the joints and fissures, with the reaction between HCl and $CaCO_3$ from marble the pH of the fluid increases. As mentioned by Liu et al. [7], cassiterite-quartz veins frequently are lined with coarse muscovite selvages, it is likely that the alteration of feldspar to muscovite in granitic rocks also contributes to the increasing in pH. Furthermore, mixing of ore fluids with dilute meteoric waters was shown to play a role in the Xuebaoding deposit, which could cause the decreasing chloride molality in fluids [7]. Changes above would cause the destroying of Sn-bearing complexes, the decreasing of $SnO_2$ solubility, and finally the large scale precipitation of Sn-bearing minerals (mainly cassiterite).

In Wood and Samson [59], the occurrence of W in high temperature hydrothermal system in different environments was compared in detail. Wood and Samson [59] proposed: (1) tungstate forms mainly as $H_2WO_4$, $HWO_4^-$, $WO_4^{2-}$, $NaWO_4^-$, $Na_2WO_4$ in fluid; tungsten complexes (-chloride, -fluoride, -carbonate complexes or more exotic species) are not necessary to form an tungsten deposit; (2) the tungsten concentration in equilibrium with scheelite increases strongly with increasing temperature, NaCl concentration and pH value; (3) simple cooling of a solution with a constant Ca/Fe ratio cannot result in the precipitation of scheelite, it requires an increasing in the Ca/Fe ratio concomitant with cooling.

As discussed above, because of the low Ca value of fluids which could improve the solubility of tungsten, mineralization of scheelite does not happen in late-magmatic stage. Then, with undercooled fluids flowing into triassic strata, content of Ca and pH increase during the hydrothermal evolution stage. No mafic minerals are found in the veins except tourmaline. Thus, it is likely that the increasing Mg and Fe mainly supplies the crystallization of tourmaline. Together with the decreasing Na values, finally caused the mineralization of W-bearing minerals which is mainly scheelite in Xuebaoding deposit.

In addition, as proposed by Pirajno and Smithies [60], the FeO/(FeO + MgO) ratio of tourmaline could be a useful indicator of spatial variations in granite-related hydrothermal W-Sn deposits hosted in siliclastic metasedimentary rocks. Systematic variations of the FeO/(FeO + MgO) ratio are observed from endogranitic deposits to distal vein systems emplaced at some distance from the granitic source. In Figure 2, the Pankou and Pukouling granites are closely related to Triassic metasedimentary strata. The higher Mg and Fe compositions of type III tourmaline, compared with type I and type II tourmalines, indicate that the biotite schist does contribute to the Xuebaoding deposit. With the FeO/(FeO + MgO) ratios ranging from 0.8 and 0.6, all types of tourmalines plot into the proximal to intermediate field. However, although such a result can not show the specific distance between the ore body and granites, it could be used as a reference suggesting that tourmalines formed as a result of fluid flow a distance from the intrusion (Figure 8).

It is proposed that the location in the country rocks where type I, type II and type III tourmalines precipitated should indicate also the presence of W-Sn-Be mineralization. Furthermore, as discussed above, in a low salinity and gradually cooling fluid, Ca content and pH could be the main variables controlling the solubility of W. The Ca and pH values would increase in the fluid due to the continuous reaction with Ca-rich sedimentary rocks. Thus, W-bearing minerals are expected to possibly extend considerably into marble and other Ca-rich sedimentary rock.

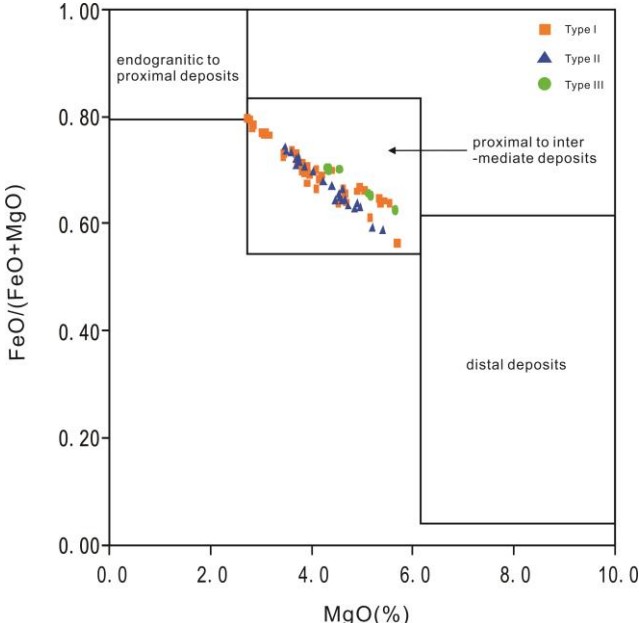

**Figure 8.** Plot of MgO vs. FeO/(FeO + MgO) of type I and type II tourmalines (after reference [60]).

## 6. Conclusions

(1) Three types of tourmalines are identified in the Xuebaoding deposit. Type I tourmalines are interpreted to have formed from an immiscible boron-rich magmatic-hydrothermal fluid. Type II tourmalines with skeletal texture formed earlier than type I, in a transition from late-magmatic to early-hydrothermal conditions, and type III are hydrothermal tourmalines, occurring in the mineralized veins.

(2) The chemical compositions of tourmaline are buffered by the host rocks. Inferred increasing $Fe^{3+}/Fe^{2+}$ ratios and the decreasing Na values of all tourmalines studied suggest that they precipitated from oxidized, low-salinity fluids.

(3) Mineralogical characteristics and chemical composition variations of tourmalines as established in this work may help in W-mineralization exploration in the larger region around Pinguw-Xuebaoding, or more generally in related geological settings.

**Author Contributions:** Conceptualization, X.Z. and Y.L.; data curation, X.Z.; writing—original draft preparation, X.Z.; writing—review and editing, M.B.R. and Y.L. All authors have read and agreed to the published version of the manuscript.

**Funding:** This study was supported by the Strategic Priority Research Program of the Chinese Academy of Sciences (Grant No. XDA20070304), the National Science Fund for Excellent Young Scholars (41922014), National Natural Science Foundation of China (Grant No. 41702096, 41772044), Fundamental Research Funds of the Chinese Academy of Geological Sciences (Grant No. YYWF201705), and a Geological Survey Program of the China Geological Survey, Ministry of Natural Resources (Grant No. DD20190060), Key Special Project for Introduced Talents Team of Southern Marine Science and Engineering Guangdong Laboratory (Guangzhou) (GML2019ZD0106).

**Acknowledgments:** We acknowledges support for field work by P. Wang, Z. Li, and former miners at Xuebaoding, and financial support from University of Electronic Science and Technology of China.

**Conflicts of Interest:** The authors declare no conflict of interest.

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
