# Peer review of "Tourmaline as a Recorder of Ore-Forming Processes in the Xuebaoding W-Sn-Be Deposit, Sichuan Province, China: Evidence from the Chemical Composition of Tourmaline"

_minerals, doi:10.3390/min10050438_

Round 1

Reviewer 1 Report

All comments are shown at the manuscript. Some significant comments are presented here:

  1. Abstract is too large.
  2. In the paper three types of tourmaline are described, but only two types are studied. Needle-like tourmaline clusters remained unexplored. Whereas this type represents tourmaline which obviously formed by hydrothermal fluid.
  3. It is necessary to add the mineral formation sequence to determine of each studied mineral position during the ore formation processes.
  4. What minerals replace tourmaline in skeletal crystals?
  5. Authors suggest that Type II tourmaline has a magmatic genesis. But simultaneously this mineral occurs in the vein selvage which exclude of its magmatic origin.
  6. Based on the above comments authors have to correct the model of the tourmaline genesis.

Author Response

Point 1. Abstract is too large.

Response 1. We have revised and shortened the abstract as recommended.

Point 2. In the paper three types of tourmaline are described, but only two types are studied. Needle-like tourmaline clusters remained unexplored. Whereas this type represents tourmaline which obviously formed by hydrothermal fluid.

Response 2. Description and chemical composition of a third needle-like tourmaline type have been added in the manuscript. This tourmaline crystals occurring in the mineralized veins are classified as type III in the reedited manuscript.

Point 3. It is necessary to add the mineral formation sequence to determine of each studied mineral position during the ore formation processes.

Response 3. As suggested we have added the mineral formation sequence.

Point 4. What minerals replace tourmaline in skeletal crystals?

Response 4. Space between the arms of the skeleton is mainly filled by albite, metasomatism residual K-feldspar and fine-grain muscovite. This description has been added in the manuscript as suggested.

Point 5. Authors suggest that Type II tourmaline has a magmatic genesis. But simultaneously this mineral occurs in the vein selvage which exclude of its magmatic origin.

Response 5. We agree and apologize for this confusion. We mean to say that the type II tourmalines are primary magmatic and derived from a highly evolved magmatic fluid crystallizing in the immediate contact of the granite to the metamorphic host. They are chemically very similar to type I tourmalines that form in the margin of the granite, yet distinct in habit and other details. E.g., it is the unique skeleton structure of type II tourmalines why we believe they can be classified as magmatic. This type of tourmaline grows along the margin of granite dikes, yet separated by a muscovite layers from the wall to the host rock. In addition, undercooled melts nucleate crystals, but if the undercooling is high enough, the crystals will have a skeletal habit. And the occurrence of skeletal crystals, is cited as evidence of undercooled crystallization of magma. Together with the no-zoned texture and the chemical compositions of type II tourmaline. This is why we suggest that type II tourmaline has a magmatic genesis.

Point 6. Based on the above comments authors have to correct the model of the tourmaline genesis.

Response 6. We appreciate these comments and have revised the manuscript accordingly.

Reviewer 2 Report

General comments:

In my opinion, the text is lacking of a detailed description of the occurrence of the two types of tourmaline. I tried to read other articles about the deposit to understand this, but I could not found a good description (in English of course, I do not known if there is one in Chinese). It is important because if the origin of Type II tourmaline is by crystallization from a melt, as the authors suggest, the ore veins would be pegmatoid veins, I means.

In the case of tourmaline Type-I, the authors suggest an hydrothermal origin, but I can not understand where this tourmaline is located in the deposit. Is it located in the border facies of the granite, along the contact between the granite and host-rocks? This is confused. Moreover, such amount of tourmaline imply a lot of Fe and Mg, and, as the authors indicate, the granite is a highly differentiated leucogranite. Does the iron come from the host-rocks? There are biotite schists in the host rocks, apart from marbles.

In the analytical methods, the authors mention “Microscope observation”, but they do not include these observations in the results. I think that it should be included, and also microphotographs, with complete description of mineral assemblages and textures. The 6 BSE images included in the text are quite similar and do not give much information.

In the case of EMPA, the authors do not indicate if fluorine was analysed. In the fluid inclusion study (Liu et al., 2012) they indicate the presence of F in the fluids. I am not an expertise in tourmaline and I suppose the formula calculations are correct, but the fluorine should be included. Moreover, I consider that the authors should mention something about the classification and nomenclature used, including references.

In the case of the diagrams where the EMPA data are plotting, there is a dispersion in the analyses of Type-I tourmaline, but the authors do not explain why. Moreover, they do not say why the tourmaline crystals appear zoned. Are the analyses of core and rim different? 

  • In figure 6, the diagram a and b should be removed and included in other figure later in the text, because they are not mentioned until line 330. In addition to this, I am not sure if these figures are necessary, because they do not give any extra information.
  • figure 6d is wrong: chorlo end-member is rich in Fe.
  • In the diagram 7a the plotting of type-I data draw two different slopes indicating two different vector of substitution, but the authors do not explain this.
  •  

The Discussion section needs to be improved. There are many repetitions, and some part of the text are confuse.

There are more comments in the pdf file.

Author Response

Point 1. In my opinion, the text is lacking of a detailed description of the occurrence of the two types of tourmaline. I tried to read other articles about the deposit to understand this, but I could not found a good description (in English of course, I do not known if there is one in Chinese). It is important because if the origin of Type II tourmaline is by crystallization from a melt, as the authors suggest, the ore veins would be pegmatoid veins, I means.

Response 1. We thank the reviewer for these valuable comments on this manuscript. We have added a more detailed descriptions of the occurrence of type I, II and the newly added type III tourmalines to the manuscript. We apologize for the confusion we have caused with regards to the type II tourmalines and their occurrence along the margin of ore veins’. This confused description has been corrected as ‘occurring along the margin of granite dikes’. Type II tourmaline always occurs along the margin of granite dikes, separated by muscovite layers from the wall rock. In addition, this type of tourmaline has unique skeletal texture. Usually, undercooled melts nucleate crystals, but if the undercooling is high enough, the crystals will have a skeletal habit. And the occurrence of skeletal crystals, is cited as evidence of undercooled crystallization of magma. Together with the no-zoned texture and the chemical compositions of type II tourmaline, it is likely that this type of tourmaline has primary magmatic origin, derived from a highly evolved magmatic fluid. Previous studies show that the Xuebaoding deposit is a hydrothermal deposit. Type II tourmaline clusters are always overgrown by muscovite layers, which separate the tourmaline clusters from mineralized veins. Therefore, the magmatic origin of type II tourmaline does not conflict with the hydrothermal origin of ore veins.

Point 2. In the case of tourmaline Type-I, the authors suggest an hydrothermal origin, but I can not understand where this tourmaline is located in the deposit. Is it located in the border facies of the granite, along the contact between the granite and host-rocks? This is confused. Moreover, such amount of tourmaline imply a lot of Fe and Mg, and, as the authors indicate, the granite is a highly differentiated leucogranite. Does the iron come from the host-rocks? There are biotite schists in the host rocks, apart from marbles.

Response 2. We apologize for the confusion. In fact, type I tourmalines occur within the granitic rocks, near the contact to dikes and veins. Immiscibility of the Pankou and Pukouling melt has been shown by Yan Liu in previous study, together with the poorly zoned texture and chemical compositions of type I tourmaline, we suggest a hydrothermal genesis of this type of tourmaline.

Although the leucogranite contains lower mafic components, it is not completely devoid of them. Generally, a concentric granitic pluton is composed by numerous rock types, and it mostly shows a variation from outer mafic to inner felsic. Type I tourmaline occurs along the margin of the border facies of the Pankou and Pukouling granites, in addition, tourmaline is the only mafic mineral found in the border facies of granite. Besides, the (Mg+Fe) values (<3 apfu) of type I and II tourmalines is a relatively low value for dravite-schorl species. Hence, it is likely that Mg and Fe of type I and II tourmalines come from its host-rock.

Biotite schists does contribute to Xuebaoding deposit, but it is mainly in hydrothermal stage. Chemical compositions of the newly added type III tourmaline show higher (Mg+Fe) values (>3 apfu), as may be caused by the reaction between hydrothermal fluid and the triassic strata,including biotite schists.

Point 3. In the analytical methods, the authors mention “Microscope observation”, but they do not include these observations in the results. I think that it should be included, and also microphotographs, with complete description of mineral assemblages and textures. The 6 BSE images included in the text are quite similar and do not give much information.

Response 3. According to your comments, the manuscript has been modified appropriately. We agree that 6 BSE images are similar, but due to the lack of electron microscope photos, we need more BSE images to prove that our description is universal. Detailed descriptions about the zonation texture, mineral assemblage and crystal shape have been added based on the BSE images.

Point 4. In the case of EMPA, the authors do not indicate if fluorine was analysed. In the fluid inclusion study (Liu et al., 2012) they indicate the presence of F in the fluids. I am not an expertise in tourmaline and I suppose the formula calculations are correct, but the fluorine should be included. Moreover, I consider that the authors should mention something about the classification and nomenclature used, including references.

Response 4. F content was not analyzed in EMPA, and quantitative EMPA of F is notoriously difficult, because of its beam sensitivity. Further it had little effect on the nomenclature of tourmaline. Relevant corresponding references have been added to the reedited manuscript.

Point 5. In the case of the diagrams where the EMPA data are plotting, there is a dispersion in the analyses of Type-I tourmaline, but the authors do not explain why. Moreover, they do not say why the tourmaline crystals appear zoned. Are the analyses of core and rim different? 

Response 5. Although occurring in granitic rocks, type I tourmaline shows only little zonation. Together with its petrographic characteristics and chemical compositions, it is likely that type I tourmaline forms from the immiscible fluid, which is a transition stage from magmatic to hydrothermal. The weak zoning corresponds to slight variations in Ca, Ti, Fe, Mg, and X-site vacancies and decreasing Al towards the rims. Descriptions of type I tourmaline have been added in the manuscript.

Point 6. In figure 6, the diagram a and b should be removed and included in other figure later in the text, because they are not mentioned until line 330. In addition to this, I am not sure if these figures are necessary, because they do not give any extra information.figure 6d is wrong: chorlo end-member is rich in Fe.

Response 6. Figure 6a and b could not only show some characteristics of the host rocks corresponding to tourmaline, but also show the proportion of Ca, Al, Mg and Fe in tourmaline. Thus they are necessary here to show the variations of Ca, Al, Mg and Fe in tourmaline. Relevant descriptions have been added and diagrams have also been revised.

Point 7. In the diagram 7a the plotting of type-I data draw two different slopes indicating two different vector of substitution, but the authors do not explain this.

Response 7. We thank the reviewer for pointing out this aspect. We agree that these tow slopes do mean two major vectors ❑Al(NaMg)−1 and MgFe2+-1. We have revised the manuscript accordingly also adding new data of type III tourmaline, and expanded the discussion.

Point 8. The Discussion section needs to be improved. There are many repetitions, and some part of the text are confuse.

Response 8. The manuscript, especially the discussion has been reedited according to this and other comments.

Reviewer 3 Report

Dear Editor,

Dear authors,

This manuscript presents a study of tourmaline from a granite-related W-Sn-Be deposit in China. The authors conclude to a magmatic-hydrothermal origin of tourmaline based on textural and chemical composition data. There are more and more studies using tourmaline as a tracer of ore-forming processes today, therefore, this manuscript may be of interest for the community of economic geologists.

I have several critics concerning the interpretation of the textures and chemical compositions of tourmaline to discuss its magmatic vs hydrothermal origin. The most important issues you should address are:

  • Textures of tourmaline: You distinguish two types of tourmaline: Type I disseminated in the granites, and Type II in mineralized veins. Based on your descriptions, it is hard to understand why you interpret Type I tourmaline as formed by a magmatic-hydrothermal fluid and Type II tourmaline as precipitated from a boron-rich silicate melt. If Type I tourmaline represents an alteration phase of the granite, you should better describe the textural observations to claim that. You describe a “skeletal structure” for Type II tourmaline, that you interpret as a proof for a magmatic origin, but then it becomes unclear if what you describe as “ore veins” represent hydrothermal veins or granitic dykes. A better description of the occurrence of these two types of tourmaline, as well as their relationship with the host (granite vs veins), is required. You distinguish several “parts” for the ore veins that you classify as Part I, II and III based on their spatial position relative to the intrusion. In the Pdf file, I suggested to replace everywhere “Part” by “Type” to clarify the descriptions. You should better describe the mineral assemblages of these veins and their relationships with the granite. Mineralogical and textural observations are fundamental to propose an interpretation of the chemical data and a formation scenario for the studied deposit.

  • Chemical composition of tourmaline: In the discussion, you try to distinguish Types I and II tourmaline based on their chemical composition measured by electron microprobe. Specifically, you make assumptions on fluid salinity based on distinct Na content of tourmaline. But looking at your diagrams all tourmaline compositions overlap and it is not possible to distinguish two populations. A possible interpretation is that Type I and Type II tourmaline formed during a single fluid event. Different textural features do not necessarily imply distinct generations of tourmaline. The mineral textures reflect the nature of the host rock, fluid interactions, and the physicochemical conditions (T-P fluctuations, etc) during precipitation of tourmaline. This should be discussed in greater detail in the text.

  • Tourmaline as indicator of W-Sn-Be mineralization: you report your data in the FeO/(FeO+MgO) vs MgO diagram of Pirajno and Smithies (1992) and propose tourmaline as a “useful exploration tool to target W-Sn-Be mineralization”. However, this diagram was originally proposed for granite-related W-Sn vein systems hosted in siliclastic metasedimentary rocks and not in marbles. You should mention this and discuss in greater detail the role of interactions between the magmatic-hydrothermal fluids and the host marbles. This section of the discussion is mostly a compilation of literature data about solubility and transport of W and Sn in hydrothermal fluids. This part should be reduced to better discuss your tourmaline data. The is some contradiction in the text because you conclude that beryl crystallize during the late-magmatic stage and then you propose that comparably to Be, W and Sn formed during a late hydrothermal stage. There are here some issues of wording and consistency with your sample descriptions that you should address.

In addition, the quality of the English is poor and requires an extensive editing. In some parts of the manuscript, it is hard to understand what the authors want to say. I highly recommend that the authors find a native English speaker to proofread the revised version of the manuscript before submission. I did numerous edits in the pdf to improve the quality of the text. The figures also require corrections in order to improve the readability of the maps and diagrams.

In my opinion and based on the presented data, the manuscript requires a major revision.

I have attached your manuscript with edits, comments and suggestions to improve the text and the discussion of your data.

Good luck with the revision.

Sincerely.

Author Response

Point 1. Textures of tourmaline: You distinguish two types of tourmaline: Type I disseminated in the granites, and Type II in mineralized veins. Based on your descriptions, it is hard to understand why you interpret Type I tourmaline as formed by a magmatic-hydrothermal fluid and Type II tourmaline as precipitated from a boron-rich silicate melt. If Type I tourmaline represents an alteration phase of the granite, you should better describe the textural observations to claim that. You describe a “skeletal structure” for Type II tourmaline, that you interpret as a proof for a magmatic origin, but then it becomes unclear if what you describe as “ore veins” represent hydrothermal veins or granitic dykes. A better description of the occurrence of these two types of tourmaline, as well as their relationship with the host (granite vs veins), is required. You distinguish several “parts” for the ore veins that you classify as Part I, II and III based on their spatial position relative to the intrusion. In the Pdf file, I suggested to replace everywhere “Part” by “Type” to clarify the descriptions. You should better describe the mineral assemblages of these veins and their relationships with the granite. Mineralogical and textural observations are fundamental to propose an interpretation of the chemical data and a formation scenario for the studied deposit.

Response 1. We thank the reviewer for these valuable comments. Descriptions of type I, II and the newly added III tourmalines have been reedited in the manuscript according to this comment. Type I tourmaline are primarily confined to the granite itself occurring at and near its border,; type II tourmaline has been redefined, and are the tourmalines in clusters in a narrow region at the contact of the granite to the metamorphic rock units, and also of primary magmatic origin; and the newly added type III tourmaline is found in the mineralized veins within the metamorphic strata. In addition, we have renamed ‘part’ to ‘zone’ as this is more estbalished terminology.

Point 2. Chemical composition of tourmaline: In the discussion, you try to distinguish Types I and II tourmaline based on their chemical composition measured by electron microprobe. Specifically, you make assumptions on fluid salinity based on distinct Na content of tourmaline. But looking at your diagrams all tourmaline compositions overlap and it is not possible to distinguish two populations. A possible interpretation is that Type I and Type II tourmaline formed during a single fluid event. Different textural features do not necessarily imply distinct generations of tourmaline. The mineral textures reflect the nature of the host rock, fluid interactions, and the physicochemical conditions (T-P fluctuations, etc) during precipitation of tourmaline. This should be discussed in greater detail in the text.

Response 2. We generally agree with this interpretation. Indeed, as discussed in the revised manuscript we believe that type II tourmaline are primary magmatic and form directly from the late magmatic fluid at the granite – metamorphic contact, type I tourmaline forms from within the granite at its border from the immiscible fluid phase separated from the parental magma. Both show similar chemical signatures, and hence it is likely that both are buffered by their close proximity to their same types of host-rocks. We now also added data on the compositions of type III tourmaline. Type III tourmaline, also chemically distinct, occurs in the mineralized veins hosted inside the metamorphic strata. It contents a relatively higher Mg+Fe values while in a lower Na and Al. The compositional trend of all types tourmaline becomes more obvious with the addition of data of type III tourmaline. We have revised and expanded result and discussion section of the manuscript accordingly.

Point 3. Tourmaline as indicator of W-Sn-Be mineralization: you report your data in the FeO/(FeO+MgO) vs MgO diagram of Pirajno and Smithies (1992) and propose tourmaline as a “useful exploration tool to target W-Sn-Be mineralization”. However, this diagram was originally proposed for granite-related W-Sn vein systems hosted in siliclastic metasedimentary rocks and not in marbles. You should mention this and discuss in greater detail the role of interactions between the magmatic-hydrothermal fluids and the host marbles. This section of the discussion is mostly a compilation of literature data about solubility and transport of W and Sn in hydrothermal fluids. This part should be reduced to better discuss your tourmaline data. The is some contradiction in the text because you conclude that beryl crystallize during the late-magmatic stage and then you propose that comparably to Be, W and Sn formed during a late hydrothermal stage. There are here some issues of wording and consistency with your sample descriptions that you should address.

Response 3. We appreciate these comments and revised the manuscript accordingly.  In addition, chemical compositions of the newly added type III tourmaline plot at the field ‘Fe3+-rich quartz-tourmaline rocks’. Mg+Fe values of type III tourmalines increase obviously. The Mg and Fe may not be derived from the granitic magma, because they are highly fractionated lecogranites. Instead we believe the that the biotite schist and fluid interaction with these hosting strata contributed as a source. Generally, a concentric granitic pluton is composed by numerous rock types, and it mostly shows a variation from outer mafic to inner felsic. Tourmalines hosted by highly fractionated granites also show early Mg-, Fe-bearing to later Al-bearing. Type III tourmaline forms from the hydrothermal fluid, as it is found in the mineralized veins, later than type I and II tourmalines. Further, data of type III tourmaline overlap with type I and II tourmalines in the FeO/(FeO+MgO) vs. MgO diagram. Hence, the FeO/(FeO+MgO) vs MgO diagram has a indicating significance in this paper.

Point 4. In addition, the quality of the English is poor and requires an extensive editing. In some parts of the manuscript, it is hard to understand what the authors want to say. I highly recommend that the authors find a native English speaker to proofread the revised version of the manuscript before submission. I did numerous edits in the pdf to improve the quality of the text. The figures also require corrections in order to improve the readability of the maps and diagrams.

Response 4. We appreciate the helpful comments. The paper has been carefully revised and type-edited to improve the language.

Round 2

Reviewer 3 Report

Dear Editor,

Dear Authors,

I have read the revised manuscript of Zhu et al and I have the same critics than in my previous review report. Although the authors have made significant efforts to improve the quality of the text and of some figures, I still consider that the manuscript suffers from several issues listed below:

1) It is still unclear for me if the Type II tourmaline are hosted in hydrothermal veins or in granitic dykes. Looking at some literature on the Xuebaoding deposit (e.g., Zhang et al. 2014, Ore Geology Reviews, v. 62, p. 315-322), this deposit is described as a pegmatite-type Sn-W-Be deposit. However, it is nowhere mentioned of pegmatites. The authors indicate that the mineralization is hosted in "ore veins", which for an economic geologist clearly mean hydrothermal veins. But the textures of the samples and the large size of the crystals suggest that the beryl-cassiterite mineralization is hosted in pegmatite dykes. It becomes more unclear when the authors discuss about exsolution of magmatic-hydrothermal fluid but then interpret the Type II tourmaline as formed by undercooling of a granitic melt. Consequently, a major effort need to be done to clarify the observations for the reader and to support the interpretation. In the present state, the paper is really hard to follow and cannot be published in my opinion because the interpretation is not supported by the observations and results.

2) In this revised version of the manuscript, the authors have completed their dataset with a third type of tourmaline (Type III) occurring in mineralized veins, therefore, of hydrothermal origin. However, the spatial and temporal relationships between these veins and the "ore veins" and granites are unclear. It is necessary to complete the field observations and sample descriptions to better understand the evolution of the deposit. In addition, I have found a major issue in the EMPA data of Type III tourmaline shown in Table 3. The SiO2 contents of tourmaline is anormally high (43-44wt%) resulting in analytical total close to 100%. This is not possible for tourmaline for which the SiO2 content is usually around 35-37wt.% and the total around 85-87%. The Al2O3 content seem also anormally elevated. I suspect here a problem of EMPA standardization. Moreover, there are inconsistencies in the structural formula calculation of Type III tourmaline. Therefore, I ask the authors to check their original data and to eventually proceed to new EMPA measurements in order to fix these issues. In consequence, the discussion of the composition of type III tourmaline cannot be relevant and will need to be revised once the problem of structural formula will have been fixed.

3) Finally, I don't see in what the chemical composition of tourmaline from Xuebaoding has implications for the W-Sn-Be mineralization. The use of the FeO/(FeO+MgO) vs. MgO diagram of Pirajno and Smithies (1992) just indicates that the tourmaline formed at a proximal distance from the intrusion. But this was already obvious from the field observations. This part of the discussion is just a summary of different results from the literature concerning the solubility of Sn, W, and Be in hydrothermal fluids. There is no reference to the previous papers on the Xuebaoding deposit and how this study inserts compared to previous works. This section is also confusing because the authors mix the evolution of magmatic fluids and melt to interpret the genesis of tourmaline. The Fe and Mg contents of tourmaline reflect the source of the fluid and the interaction with the host rocks, but do not necessarily indicate the existence of a Sn-W-Be mineralization (especially if that one is hosted in pegmatites). To claim this, you would need to know the Sn-W-Be contents of tourmaline. There are plenty of occurrences of dravite-schorl tourmaline worldwide without associated mineralization, so using the FeO/MgO ratio of tourmaline as a proxy for mineralization is not reliable. I consider that this part of the discussion need to be reworked once the main issues above will have been fixed.

I have attached the pdf file with some edits and comments. The text requires again some editing of the English language and style and some figures should be modified as well. In my opinion, the manuscript requires a major revision.

Good luck with the revision.

Best regards

Author Response

Point 1. It is still unclear for me if the Type II tourmaline are hosted in hydrothermal veins or in granitic dykes. Looking at some literature on the Xuebaoding deposit (e.g., Zhang et al. 2014, Ore Geology Reviews, v. 62, p. 315-322), this deposit is described as a pegmatite-type Sn-W-Be deposit. However, it is nowhere mentioned of pegmatites. The authors indicate that the mineralization is hosted in "ore veins", which for an economic geologist clearly mean hydrothermal veins. But the textures of the samples and the large size of the crystals suggest that the beryl-cassiterite mineralization is hosted in pegmatite dykes. It becomes more unclear when the authors discuss about exsolution of magmatic-hydrothermal fluid but then interpret the Type II tourmaline as formed by undercooling of a granitic melt. Consequently, a major effort need to be done to clarify the observations for the reader and to support the interpretation. In the present state, the paper is really hard to follow and cannot be published in my opinion because the interpretation is not supported by the observations and results.

Response 1. We apologize for the confusions. Type II tourmaline grows along the margin of the granites, the word ‘dikes’ has been deleted in our manuscript. We read the reference Zhang et al. 2014, there may be a confusion in terminology. No zonal structure was found within the veins, even the granite-host veins, there is only a layer of muscovite separating coarse mineral and host rock, hence we consider that the Xuebaoding deposit is a typical hydrothermal deposit which has been proved by Liu et al. (references 4-7). In addition, only type III tourmalines could be found coexisting with coarse-grained scheelite, beryl, cassiterite and other coarse-grained minerals.

Point 2. In this revised version of the manuscript, the authors have completed their dataset with a third type of tourmaline (Type III) occurring in mineralized veins, therefore, of hydrothermal origin. However, the spatial and temporal relationships between these veins and the "ore veins" and granites are unclear. It is necessary to complete the field observations and sample descriptions to better understand the evolution of the deposit. In addition, I have found a major issue in the EMPA data of Type III tourmaline shown in Table 3. The SiO2 contents of tourmaline is anormally high (43-44wt%) resulting in analytical total close to 100%. This is not possible for tourmaline for which the SiO2 content is usually around 35-37wt.% and the total around 85-87%. The Al2O3 content seem also anormally elevated. I suspect here a problem of EMPA standardization. Moreover, there are inconsistencies in the structural formula calculation of Type III tourmaline. Therefore, I ask the authors to check their original data and to eventually proceed to new EMPA measurements in order to fix these issues. In consequence, the discussion of the composition of type III tourmaline cannot be relevant and will need to be revised once the problem of structural formula will have been fixed.

Response 2. As mentioned in Point 1, relationships between these veins and the "ore veins" and granites have been corrected. As a hydrothermal deposit, veins in the Xuebaoding deposit are all W-Sn-Be mineralized. There are veins in granite, some of which continue into the schist, others are spatially offset, but they have the respresentative types of I, II, and III tourmaline. We apologize for issue you found in the EMPA data of type III tourmaline, it is because that we have normalized it, and we have now replaced it with the original data. However, structural formula of tourmaline needs to be recalculated on the basis of a total of 15 cations in the octahedral and tetrahedral, this is actually a proportional relationship, and whether EMPA has been normalized has no effect on the recalculated type III tourmaline composition. Thus, the chemical compositions and figures of the type III tourmaline has not been changed after our recalculation.

Point 3. Finally, I don't see in what the chemical composition of tourmaline from Xuebaoding has implications for the W-Sn-Be mineralization. The use of the FeO/(FeO+MgO) vs. MgO diagram of Pirajno and Smithies (1992) just indicates that the tourmaline formed at a proximal distance from the intrusion. But this was already obvious from the field observations. This part of the discussion is just a summary of different results from the literature concerning the solubility of Sn, W, and Be in hydrothermal fluids. There is no reference to the previous papers on the Xuebaoding deposit and how this study inserts compared to previous works. This section is also confusing because the authors mix the evolution of magmatic fluids and melt to interpret the genesis of tourmaline. The Fe and Mg contents of tourmaline reflect the source of the fluid and the interaction with the host rocks, but do not necessarily indicate the existence of a Sn-W-Be mineralization (especially if that one is hosted in pegmatites). To claim this, you would need to know the Sn-W-Be contents of tourmaline. There are plenty of occurrences of dravite-schorl tourmaline worldwide without associated mineralization, so using the FeO/MgO ratio of tourmaline as a proxy for mineralization is not reliable. I consider that this part of the discussion need to be reworked once the main issues above will have been fixed.

Response 3. We accept your comments, and revised the section 5.3, the Xuebaoding deposit has been proved to be a hydrothermal deposit by previous studies. Therefore, we would like to compare the differences between the magmatic and hydrothermal stage of the fluid, so as to explain why minerals crystallize mainly in the hydrothermal stage in the Xuebaoding deposit. The FeO/MgO ratio here is an auxiliary reference, which indicates that the ore-forming elements does migrate in a distance and mineralization does not take place in the granite body.